# Intrinsic Motivation via Surprise Memory

## Abstract

We present a new computing model for intrinsic rewards in reinforcement learning that addresses the limitations of existing surprise-driven explorations. The reward is the *novelty of the surprise* rather than the surprise norm. We estimate the surprise novelty as retrieval errors of a memory network wherein the memory stores and reconstructs surprises. Our surprise memory (SM) augments the capability of surprise-based intrinsic motivators, maintaining the agent's interest in exciting exploration while reducing unwanted attraction to unpredictable or noisy observations. Our experiments demonstrate that the SM combined with various surprise predictors exhibits efficient exploring behaviors and significantly boosts the final performance in sparse reward environments, including Noisy-TV, navigation and challenging Atari games.

## 1 Introduction

*What motivates agents to explore?* Successfully answering this question would enable agents to learn efficiently in formidable tasks. Random explorations such as $\epsilon$-greedy are inefficient in high dimensional cases, failing to learn despite training for hundreds of million steps in sparse reward games (Bellemare et al., 2016). Alternative approaches propose to use intrinsic motivation to aid exploration by adding bonuses to the environment's rewards (Bellemare et al., 2016; Stadie et al., 2015). The intrinsic reward is often proportional to the novelty of the visiting state: it is high if the state is novel (e.g. different from the past ones (Badia et al., 2020; 2019)) or less frequently visited (Bellemare et al., 2016; Tang et al., 2017).

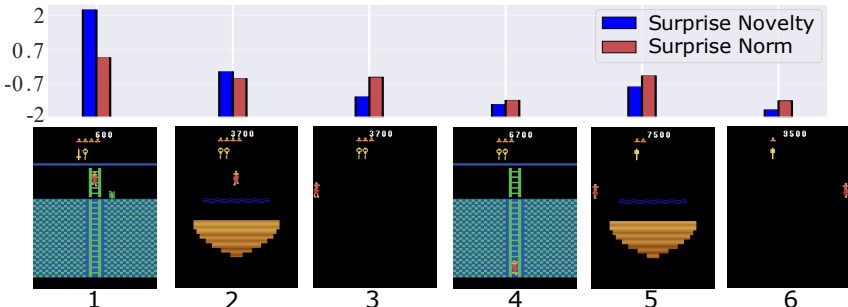

Figure 1: Montezuma Revenge: *surprise novelty* better reflects the originality of the environment than *surprise norm*. While surprise norm can be significant even for dull events such as those in the dark room due to unpredictability, surprise novelty tends to be less ($3^{rd}$ and $6^{th}$ image). On the other hand, surprise novelty can be higher in truly vivid states on the first visit to the ladder and island rooms ($1^{st}$ and $2^{nd}$ image) and reduced on the second visit ($4^{th}$ and $5^{th}$ image). Here, surprise novelty and surprise norm are quantified and averaged over steps in each room.

Another view of intrinsic motivation is from surprise, which refers to the result of the experience being unexpected, and is determined by the discrepancy between the expectation (from the gent's prediction) and observed reality (Barto et al., 2013; Schmidhuber, 2010). Technically, surprise is the difference between prediction and observation representation vectors. The norm of the residual (i.e. prediction error) is used as the intrinsic reward.

Here, we will use the terms "surprise" and "surprise norm" to refer to the residual vector and its norm, respectively. Recent works have estimated surprise with various predictive models such as dynamics (Stadie et al., 2015), episodic reachability (Savinov et al., 2018) and inverse dynamics (Pathak et al., 2017); and achieved significant improvements with surprise norm (Burda et al., 2018a). However, surprise-based agents tend to be overly curious about noisy or unpredictable observations (Itti and Baldi, 2005; Schmidhuber, 1991). For example, consider an agent watching a television screen showing white noise (noisy-TV problem). The TV is boring, yet the agent cannot predict the screen's content and will be attracted to the TV due to its high surprise norm. This distraction or "fake surprise" is common in partially observable Markov Decision Process (POMDP), including navigation tasks and Atari games (Burda et al., 2018b). Many works have addressed this issue by relying on the learning progress (Achiam and Sastry, 2017; Schmidhuber, 1991) or random network distillation (RND) (Burda et al., 2018b). However, the former is computationally expensive, and the latter requires many samples to perform well.

This paper overcomes the "fake surprise" issue by using *surprise novelty* - a new concept that measures the uniqueness of surprise. To identify surprise novelty, the agent needs to compare the current surprise with surprises in past encounters. One way to do this is to equip the agent with some kind of associative memory, which we implement as an autoencoder whose task is to reconstruct a query surprise. The lower the reconstruction error, the lower the surprise novelty. A further mechanism is needed to deal with the rapid changes in surprise structure within an episode. As an example, if the agent meets the same surprise at two time steps, its surprise novelty should decline, and with a simple autoencoder this will not happen. To remedy this, we add an episodic memory, which stores intra-episode surprises. Given the current surprise, this memory can retrieve similar "surprises" presented earlier in the episode through an attention mechanism. These surprises act as a context added to the query to help the autoencoder better recognize whether the query surprise has been encountered in the episode or not. The error between the query and the autoencoder's output is defined as *surprise novelty*, to which the intrinsic reward is set proportionally.

We argue that using surprise novelty as an intrinsic reward is better than surprise norm. As in POMDPs, surprise norms can be very large since the agent cannot predict its environment perfectly, yet there may exist patterns of prediction failure. If the agent can remember these patterns, it will not feel surprised when similar prediction errors appear regardless of the surprise norms. An important emergent property of this architecture is that when random observations are presented (e.g., white noise in the noisy-TV problem), the autoencoder can act as an identity transformation operator, thus effectively "passing the noise through" to reconstruct it with low error. We conjecture that the autoencoder is able to do this with the surprise rather than the observation as the surprise space has lower variance, and we show this in our paper. To make our memory system work on the surprise level, we adopt an intrinsic motivation method to generate surprise for the memory. The surprise generator (SG) can be of any kind based on predictive models and is jointly trained with the memory to optimize its own loss function. To train the surprise memory (SM), we optimize the memory's parameters to minimize the reconstruction error.

Our contribution is to propose a new concept of surprise novelty for intrinsic motivation. We argue that it reflects better the environment originality than surprise norm (see motivating graphics Fig. 1). In our experiments, the SM helps RND (Burda et al., 2018b) perform well in our challenging noisy-TV problem while RND alone performs poorly. Not only with RND, we consistently demonstrate significant performance gain when coupling three different SGs with our SM in sparse-reward tasks. Finally, in hard exploration Atari games, we boost the scores of 2 strong SGs, resulting in better performance under the low-sample regime.

## 2 METHODS

### 2.1 SURPRISE NOVELTY

Surprise is the difference between expectation and observation (Ekman and Davidson, 1994). If a surprise repeats, it is no longer a surprise. Based on this intuition, we hypothesize that surprises can be characterized by their novelties, and an agent's curiosity is driven by the

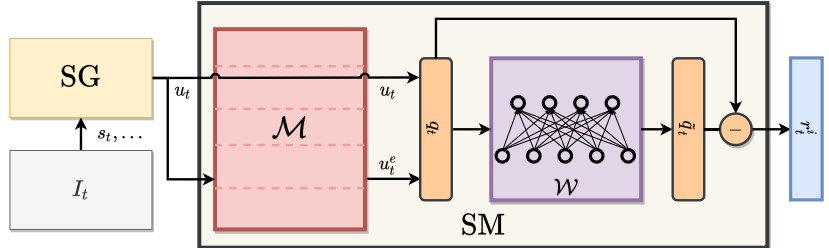

Figure 2: Surprise Generator+Surprise Memory (SG+SM). The SG takes input $I_t$ from the environment to estimate the surprise $u_t$ at state $s_t$. The SM consists of two modules: an episodic memory ($\mathcal{M}$) and an autoencoder network ($\mathcal{W}$). $\mathcal{M}$ is slot-based, storing past surprises within the episode. At any timestep $t$, given surprise $u_t$, $\mathcal{M}$ retrieves read-out $u_t^e$ to form a query surprise $q_t = [u_t^e, u_t]$ to $\mathcal{W}$. $\mathcal{W}$ tries to reconstruct the query and takes the reconstruction error (surprise novelty) as the intrinsic reward $r_t^i$.

surprise novelty rather than the surprising magnitude. Moreover, surprise novelty should be robust against noises: it is small even for random observations. For example, watching a random-channel TV can always be full of surprises as we cannot expect which channel will appear next. However, the agent should soon find it boring since the surprise of random noises reoccurs repeatedly, and the channels are entirely unpredictable.

We propose using a memory-augmented neural network (MANN) to measure surprise novelty. The memory remembers past surprise patterns, and if a surprise can be retrieved from the memory, it is not novel, and the intrinsic motivation should be small. The memory can also be viewed as a reconstruction network. The network can pass its inputs through for random, pattern-free surprises, making them retrievable. Surprise novelty has an interesting property: if some event is unsurprising (the expectation-reality residual is $\vec{0}$), its surprise ($\vec{0}$ with norm 0) is always perfectly retrievable (surprise novelty is 0). In other words, low surprise norm means low surprise novelty. On the contrary, high surprise norm can have little surprise novelty as long as the surprise can be retrieved from the memory either through associative recall or pass-through mechanism. Another property is that the variance of surprise is generally lower than that of observation (state), potentially making the learning on surprise space easier. This property is formally stated as follows.

**Proposition 1.** *Let $X$ and $U$ be random variables representing the observation and surprise at the same timestep, respectively. Under an imperfect SG, the following inequality holds:*

$$\forall i: \left(\sigma_i^X\right)^2 \geq \left(\sigma_i^U\right)^2$$

*where $\left(\sigma_i^X\right)^2$ and $\left(\sigma_i^U\right)^2$ denote the i-th diagonal elements of $\mathrm{var}(X)$ and $\mathrm{var}(U)$, respectively.*

*Proof.* See Appendix E. □

## 2.2 SURPRISE GENERATOR

Since our MANN requires surprises for its operation, it is built upon a prediction model, which will be referred to as Surprise Generators (SG). In this paper, we adopt many well-known SGs (e.g. RND (Burda et al., 2018b) and ICM (Pathak et al., 2017)) to predict the observation, compute the surprise $u_t$ and its norm for every step in the environment. The surprise norm is the Euclidean distance between the expectation and the actual observation:

$$\|u_t\| = \|SG(I_t) - O_t\| \tag{1}$$

where $u_t \in \mathbb{R}^n$ is the surprise vector of size $n$, $I_t$ the input of the SG at step $t$ of the episode, $SG(I_t)$ and $O_t$ the SG's prediction and the observation target, respectively. The input $I_t$ is specific to the SG architecture choice, which can be the current ($s_t$) or previous state, action ($s_{t-1}, a_t$). The observation target $O_t$ is usually a transformation (can be identical or

random) of the current state $s_t$, which serves as the target for the SG's prediction. The SG is usually trained to minimize:

$$\mathcal{L}_{SG} = \mathbb{E}_t \left[ \|u_t\| \right] \tag{2}$$

Here, predictable observations have minor prediction errors or little surprise. One issue is that a great surprise norm can be simply due to noisy or distractive observations. Next, we propose a remedy for this problem.

## 2.3 Surprise Memory

The surprise generated by the SG is stored and processed by a memory network dubbed Surprise Memory (SM). It consists of an episodic memory $\mathcal{M}$ and an autoencoder network $\mathcal{W}$, jointly optimized to reconstruct any surprise. At each timestep, the SM receives a surprise $u_t$ from the SG module and reads content $u_t^e$ from the memory $\mathcal{M}$. $\{u_t^e, u_t\}$ forms a surprise query $q_t$ to $\mathcal{W}$ to retrieve the reconstructed $\tilde{q}_t$. This reconstruction will be used to estimate the novelty of surprises forming intrinsic rewards $r_t^i$. Fig. 2 summarizes the operations of the components of our proposed method. Our 2 memory design effectively recovers surprise novelty by handling intra and inter-episode surprise patterns thanks to $\mathcal{M}$ and $\mathcal{W}$, respectively. $\mathcal{M}$ can quickly adapt and recall surprises that occur within an episode. $\mathcal{W}$ is slower and focuses more on consistent surprise patterns across episodes during training.

Here the query $q_t$ can be directly set to the surprise $u_t$. However, this ignores the rapid change in surprise within an episode. Without $\mathcal{M}$, when the SG and $\mathcal{W}$ are fixed (during interaction with environments), their outputs $u_t$ and $\tilde{q}_t$ stay the same for the same input $I_t$. Hence, the intrinsic reward $r_t^i$ also stays the same. It is undesirable since when the agent observes the same input at different timesteps (e.g., $I_1 = I_2$), we expect its curiosity should decrease in the second visit ($r_1^i < r_2^i$). Therefore, we design SM with $\mathcal{M}$ to fix this issue.

**The episodic memory** $\mathcal{M}$ stores representations of surprises that the agent encounters during an episode. For simplicity, $\mathcal{M}$ is implemented as a first-in-first-out queue whose size is fixed as $N$. Notably, the content of $\mathcal{M}$ is wiped out at the end of each episode. Its information is limited to a single episode. $\mathcal{M}$ can be viewed as a matrix: $\mathcal{M} \in \mathbb{R}^{N \times d}$, where $d$ is the size of the memory slot. We denote $\mathcal{M}(j)$ as the $j$-th row in the memory, corresponding to the surprise $u_{t-j}$. To retrieve from $\mathcal{M}$ a read-out $u_t^e$ that is close to $u_t$, we perform content-based attention (Graves et al., 2014) to compute the attention weight as $w_t(j) = \frac{(u_t Q)\mathcal{M}(j)^\top}{\|(u_t Q)\| \|\mathcal{M}(j)\|}$. The read-out from $\mathcal{M}$ is then $u_t^e = w_t \mathcal{M} V \in \mathbb{R}^n$. Here, $Q \in \mathbb{R}^{n \times d}$ and $V \in \mathbb{R}^{d \times n}$ are learnable weights mapping between the surprise and the memory space. To force the read-out close to $u_t$, we minimize:

$$\mathcal{L}_{\mathcal{M}} = \mathbb{E}_t \left[ \|u_t^e - u_t\| \right] \tag{3}$$

The read-out and the SG's surprise form the query surprise to $\mathcal{W}$: $q_t = [u_t^e, u_t] \in \mathbb{R}^{2n}$.

$\mathcal{M}$ stores intra-episode surprises to assist the autoencoder in preventing the agent from exploring "fake surprise" within the episode. Since we train the parameters to reconstruct $u_t$ using past surprises in the episode, if the agent visits a state whose surprise is predictable from those in $\mathcal{M}$, $\|u_t^e - u_t\|$ should be small. Hence, the read-out context $u_t^e$ contains no extra information than $u_t$ and reconstructing $q_t$ from $\mathcal{W}$ becomes easier as it is equivalent to reconstructing $u_t$. In contrast, visiting diverse states leads to a more novel read-out $u_t^e$ and makes it more challenging to reconstruct $q_t$, generally leading to higher intrinsic reward.

**The autoencoder network** $\mathcal{W}$ can be viewed as an associative memory of surprises that persist across episodes. At timestep $t$ in any episode during training, $\mathcal{W}$ is queried with $q_t$ to produce a reconstructed memory $\tilde{q}_t$. The surprise novelty is then determined as:

$$r_t^i = \|\tilde{q}_t - q_t\| \tag{4}$$

which is the norm of the surprise residual $\tilde{q}_t - q_t$. It will be normalized and added to the external reward as an intrinsic reward bonus. The details of computing and using normalized intrinsic rewards can be found in Appendix C.

We implement $\mathcal{W}$ as a feed-forward neural network that learns to reconstruct its own inputs. This kind of autoencoder has been shown to be equivalent to an associative memory that

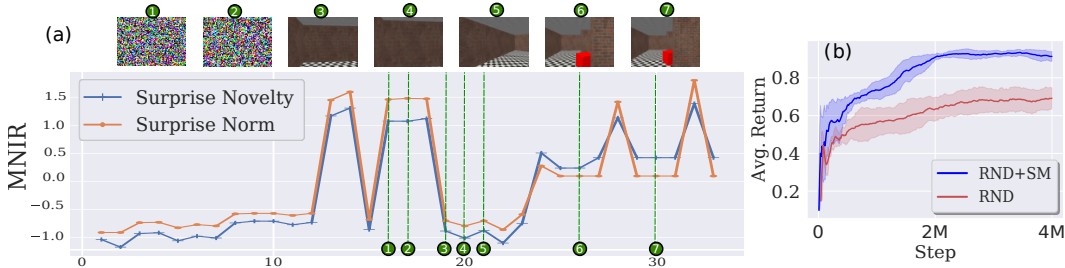

Figure 3: Noisy-TV: (a) mean-normalized intrinsic reward (MNIR) produced by RND and RND+SM at 7 selected steps in an episode. (b) Average task return (mean±std. over 5 runs) over 4 million training steps.

supports memory encoding and retrieval through attractor dynamics (Radhakrishnan et al., 2020). The query surprise is encoded to the weights of the network via backpropagation as we minimize the reconstruction loss below:

$$\mathcal{L}_{\mathcal{W}} = \mathbb{E}_t \left[ r_t^i \right] = \mathbb{E}_t \left[ \| \mathcal{W} \left( q_t \right) - q_t \| \right] \tag{5}$$

Here, $\tilde{q}_t = \mathcal{W} \left( q_t \right)$. Intuitively, it is easier to retrieve non-novel surprises experienced many times in past episodes. Thus, the intrinsic reward is lower for states that leads to these familiar surprises. On the contrary, rare surprises are harder to retrieve, which results in high reconstruction errors and intrinsic rewards. $\mathcal{W}$ is like a long-term inter-episode associative memory. Unlike slot-based memories, it has a fixed memory capacity, can compress information and learn data representations. We could store the surprise in a slot-based memory across episodes, but the size of this memory would be autonomous, and the data would be stored redundantly. Hence, the quality of the stored surprise will reduce as more and more observations come in. Readers can refer to Appendix A to see the architecture details and how $\mathcal{W}$ can be interpreted as implementing associative memory. The whole system SG+SM is trained end-to-end by minimizing the following loss: $\mathcal{L} = \mathcal{L}_{SG} + \mathcal{L}_{\mathcal{M}} + \mathcal{L}_{\mathcal{W}}$. Here, we block the gradients from $\mathcal{L}_{\mathcal{W}}$ backpropagated to the parameters of SG to avoid trivial reconstructions of $q_t$. Pseudocode of our algorithm is presented in Appendix B.

## 3 Experimental Results

### 3.1 Noisy-TV: Robustness against Noisy Observations

We use Noisy-TV, an environment designed to fool exploration methods (Burda et al., 2018b; Savinov et al., 2018), to confirm that our method can generate intrinsic rewards that (1) are more robust to noises and (2) can discriminate rare and common observations through surprise novelty. We simulate this problem by employing a 3D maze environment with a random map structure. The TV is not fixed in specific locations in the maze to make it more challenging. Instead, the agent "brings" the TV with it and can choose to watch TV anytime. Hence, there are three basic actions (turn left, right, and move forward) plus an action: watch TV. When taking this action, the agent will see a white noise image sampled from standard normal distribution and thus, the number of TV channels can be considered infinity. The agent's state is an image of its viewport, and its goal is to search for a red box randomly placed in the maze (+1 reward if the agent reaches the goal). The baseline is RND (Burda et al., 2018b), a simple yet strong SG that is claimed to obviate the stochastic problems of Noisy-TV. Our SG+SM model uses RND as the SG, so we name it RND+SM. Since our model and the baseline share the same RND architecture, the difference in performance must be attributed to our SM.

Fig. 3 (a) illustrates the mean-normalized intrinsic rewards (MNIR)[1] measured at different states in our Noisy-TV environment. The first two states are noises, the following three states are common walls, and the last two are ones where the agent sees the box. The

---

[1]See Appendix C for more information on this metric.

| Task | Baseline | RND/SM | ICM/SM | NGU/SM | AE/SM |
|------|----------|--------|--------|--------|-------|
| KD | 0.0±0.0 | 48.3±26/*79.3±4* | 5.9±5/4.7±3 | 64.4±3/*83.4±4* | 1.4±1/***91.2±6*** |
| DO | -27.0±0.7 | -13.6±8/***70.8±11*** | -27.7±2/*43.6±16* | -23.9±3/*48.6±28* | -5.1±2/*67.5±13* |
| LC | 78.0±1.7 | 25.0±35/*71.1±5* | 56.2±40/***84.6±1*** | 42.2±40/*69.5±5* | 29.0±6/*70.9±2* |

Table 1: MiniGrid: test performance after 10 million training steps. The numbers are average task return×100 over 128 episodes (mean±std. over 5 runs). Bold denotes best results on each task. Italic denotes that SG+SM is better than SG regarding Cohen effect size less than 0.5.

MNIR bars show that both models are attracted mainly by the noisy TV, resulting in the highest MNIRs. However, our model with SM suffers less from noisy TV distractions since its MNIR is lower than RND's. We speculate that SM is able to partially reconstruct the white-noise surprise via pass-through mechanism, making the normalized surprise novelty generally smaller than the normalized surprise norm in this case. That mechanism is enhanced in SM with surprise reconstruction (see Appendix D.1 for explanation).

On the other hand, when observing red box, RND+SM shows higher MNIR than RND. The difference between MNIR for common and rare states is also more prominent in RND+SM than in RND because RND prediction is not perfect even for common observations, creating relatively significant surprise norms for seeing walls. The SM fixes that issue by remembering surprise patterns and successfully retrieving them, producing much smaller surprise novelty compared to those of rare events like seeing red box. Consequently, the agent with SM outperforms the other by a massive margin in task rewards (Fig. 3 (b)).

As we visualize the number of watching TV actions and the value of the intrinsic reward by RND+SM and RND over training time, we realize that RND+SM helps the agent take fewer watching actions and thus, collect smaller amounts of intrinsic rewards compared to RND. We also verify that our proposed method outperforms a simplified version of SM using counts to measure surprise novelty and a vanilla baseline that does not use intrinsic motivation. The details of these results are given in Appendix D.1.

## 3.2 MiniGrid: Compatibility with Different Surprise Generators

We show the versatility of our framework SG+SM by applying SM to 4 SG backbones: RND (Burda et al., 2018b), ICM (Pathak et al., 2017), NGU (Badia et al., 2019) and autoencoder-AE (see Appendix D.2 for implementation details). We test the models on three tasks from MiniGrid environments: Key-Door (KD), Dynamic-Obstacles (DO) and Lava-Crossing (LC) (Chevalier-Boisvert et al., 2018). If the agent reaches the goal in the tasks, it receives a +1 reward. Otherwise, it can be punished with negative rewards if it collides with obstacles or takes too much time to finish the task. These environments are not stochastic as the Noisy-TV but they still contain other types of distraction. For example, in KD, the agent can be attracted to irrelevant actions such as going around to drop and pick the key. In DO, instead of going to the destination, the agent may chase obstacle balls flying around the map. In LC the agent can commit unsafe actions like going near lava areas, which are different from typical paths. In any case, due to reward sparsity, intrinsic motivation is beneficial. However, surprise alone may not be enough to guide an efficient exploration since the observation can be too complicated for SG to minimize its prediction error. Thus, the agent quickly feels surprised, even in unimportant states.

Table 1 shows the average returns of the models for three tasks. The Baseline is the PPO backbone trained without intrinsic reward. RND, ICM, NGU and AE are SGs providing the PPO with surprise-norm rewards while our method SG+SM uses surprise-novelty rewards. The results demonstrate that models with SM often outperform SG significantly and always contain the best performers. Notably, in the LC task, SGs hinder the performance of the Baseline because the agents are attracted to dangerous vivid states, which are hard to predict but cause the agent's death. The SM models avoid this issue and outperform the Baseline for the case of ICM+SM. Compared to AE, which computes intrinsic reward based on the novelty of the state, AE+SM shows a much higher average score in all tasks. That manifests the importance of modeling the novelty of surprise instead of states.

| Task | EMI♠ | LWM♠ | RND♠ | LWM◇ | LWM+SM◇ | RND◇ | RND+SM◇ |
|---|---|---|---|---|---|---|---|
| Freeway | **33.8** | 30.8 | 33.3 | 31.1 | 31.6 | 22.2 | 22.2 |
| Frostbite | 7,002 | 8,409 | 2,227 | 8,598 | *10,258* | 2,628 | *5,073* |
| Venture | 646 | 998 | 707 | 985 | *1,381* | 1,081 | *1,119* |
| Gravitar | 558 | 1,376 | 546 | 1,242 | *1,693* | 739 | *987* |
| Solaris | 2,688 | 1,268 | 2,051 | 1,839 | *2,065* | 2,206 | *2,420* |
| Montezuma | 387 | 2,276 | 377 | 2,192 | 2,269 | 2,475 | **5,187** |
| Norm. Mean | 61.4 | 80.6 | 42.2 | 80.5 | **97.0** | 50.7 | *74.8* |
| Norm. Median | 34.9 | 60.8 | 32.7 | 66.5 | *83.7* | 58.3 | ***84.6*** |

Table 2: Atari: average return over 128 episodes after 50 million training frames (mean over 5 runs). ♠ is from a prior work (Ermolov and Sebe, 2020). ◇ is our run. The last two rows are mean and median human normalized scores. Bold denotes best results. Italic denotes that SG+SM is significantly better than SG regarding Cohen effect size less than 0.5.

To analyze the difference between the SG+SM and SG's MNIR structure, we visualize the MNIR for each cell in the map of Key-Door in Appendix's Figs. 5 (b) and (c). We create a synthetic trajectory that scans through all the cells in the big room on the left and, at each cell, uses RND+SM and RND models to compute the corresponding surprise-norm and surprise-novelty MNIRs, respectively. As shown in Fig. 5 (b), RND+SM selectively identifies truly surprising events, where only a few cells have high surprise-novelty MNIR. Here, we can visually detect three important events that receive the most MNIR: seeing the key (bottom row), seeing the door side (in the middle of the rightmost column) and approaching the front of the door (the second and fourth rows). Other less important cells are assigned very low MNIR. On the contrary, RND often gives high surprise-norm MNIR to cells around important ones, which creates a noisy MNIR map as in Fig. 5 (c). As a result, RND's performance is better than Baseline, yet far from that of RND+SM. Another analysis of how surprise novelty discriminates against surprises with similar norms is given in Appendix's Fig. 8.

## 3.3 Atari: Sample-efficient Benchmark

We adopt the sample-efficiency Atari benchmark (Kim et al., 2019) on six hard exploration games where the training budget is only 50 million frames. We use our SM to augment 2 SGs: RND (Burda et al., 2018b) and LWM (Ermolov and Sebe, 2020). Unlike RND, LWM uses a recurrent world model and forward dynamics to generate surprises. Details of the SGs, training and evaluation are in Appendix D.3. We run the SG and SG+SM in the same codebase and setting. Table 2 reports our and representative results from prior works, showing SM-augmented models outperform their SG counterparts in all games (same codebase). In Frostbite and Montezuma Revenge, RND+SM's score is almost twice as many as that of RND. For LWM+SM, games such as Gravitar and Venture observe more than 40% improvement. Overall, LWM+SM and RND+SM achieve the best mean and median human normalized score, improving 16% and 22% w.r.t the best SGs, respectively. Notably, RND+SM shows significant improvement for the notorious Montezuma Revenge.

We also verify the benefit of the SM in the long run for Montezuma Revenge and Frostbite. As shown in Fig. 4 (a,b), RND+SM still significantly outperforms RND after 200 million training frames, achieving average scores of 10,000 and 9,000, respectively. The result demonstrates the scalability of our proposed method. When using RND and RND+SM to compute the average MNIR in several rooms in Montezuma Revenge (Fig. 1), we realize that SM makes MNIR higher for surprising events in rooms with complex structures while depressing the MNIR of fake surprises in dark rooms. Here, even in the dark room, the movement of agents (human or spider) is hard to predict, leading to a high average MNIR. On the contrary, the average MNIR of surprise novelty is reduced if the prediction error can be recalled from the memory.

Finally, measuring the running time of the models, we notice little computing overhead caused by our SM. On our Nvidia A100 GPUs, LWM and LWM+SM's average time for one 50M training are 11h 38m and 12h 10m, respectively. For one 200M training, RND and

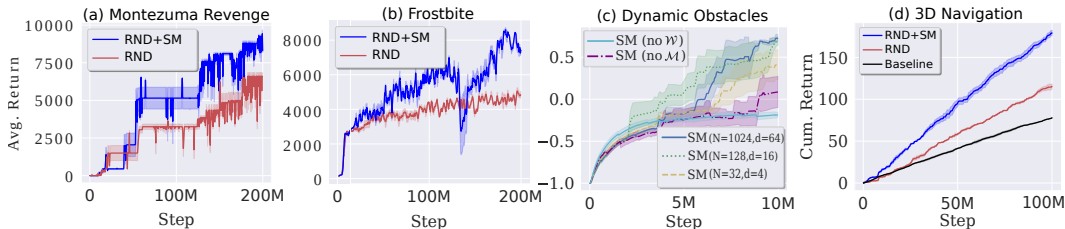

Figure 4: (a,b) Atari long runs over 200 million frames: average return over 128 episodes. (c) Ablation study on SM's components. (d) MiniWorld exploration without task reward: Cumulative task returns over 100 million training steps for the hard setting. The learning curves are mean±std. over 5 runs.

RND+SM's average times are 26h 24m and 28h 1m, respectively. These correspond to only 7% more training time while the performance gap is significant (4000 scores).

### 3.4 Ablation Study

**Role of Memories** Here, we use Minigrid's Dynamic-Obstacle task to study the role of $\mathcal{M}$ and $\mathcal{W}$ in the SM (built upon RND as the SG). Disabling $\mathcal{W}$, we directly use $\|q_t\| = \|[u_t^e, u_t]\|$ as the intrinsic reward, and name this version: SM (no $\mathcal{W}$). To ablate the effect of $\mathcal{M}$, we remove $u_t^e$ from $q_t$ and only use $q_t = u_t$ as the query to $\mathcal{W}$, forming the version: SM (no $\mathcal{M}$). We also consider different episodic memory capacity and slot size $N$-$d=$ $\{32-4, 128-16, 1024-64\}$. As $N$ and $d$ increase, the short-term context expands and more past surprise information is considered in the attention. In theory, a big $\mathcal{M}$ is helpful to capture long-term and more accurate context for constructing the surprise query.

Fig. 4 (c) depicts the performance curves of the methods after 10 million training steps. SM (no $\mathcal{W}$) and SM (no $\mathcal{M}$) show weak signs of learning, confirming the necessity of both modules in this task. Increasing $N$-$d$ from $32-4$ to $1024-64$ improves the final performance. However, $1024-64$ is not significantly better than $128-16$, perhaps because it is unlikely to have similar surprises that are more than 128 steps apart. Thus, a larger attention span does not provide a benefit. As a result, we keep using $N = 128$ and $d = 16$ in all other experiments for faster computing. We also verify the necessity of $\mathcal{M}$ and $\mathcal{W}$ in Montezuma Revenge and illustrate how $\mathcal{M}$ generates lower MNIR when 2 similar event occurs in the same episode in Key-Door (see Appendix D.4).

**No Task Reward** In this experiment, we remove task rewards and merely evaluate the agent's ability to explore using intrinsic rewards. The task is to navigate 3D rooms and get a +1 reward for picking an object (Chevalier-Boisvert, 2018). The state is the agent's image view, and there is no noise. Without task rewards, it is crucial to maintain the agent's interest in unique events of seeing the objects. In this partially observable environment, surprise-prediction methods may struggle to explore even without noise due to lacking information for good predictions, leading to usually high prediction errors. For this testbed, we evaluate random exploration agent (Baseline), RND and RND+SM in 2 settings: 1 room with three objects (easy), and 4 rooms with one object (hard).

To see the difference among the models, we compare the cumulative task rewards over 100 million steps (see Appendix D.4 for details). RND is even worse than Baseline in the easy setting because predicting causes high biases (intrinsic rewards) towards the unpredictable, hindering exploration if the map is simple. In contrast, RND+SM uses surprise novelty, generally showing smaller intrinsic rewards (see Appendix Fig. 12 (right)). Consequently, our method consistently demonstrates significant improvements over other baselines (see Fig. 4 (d) for the hard setting).

## 4 Related works

Intrinsic motivation approaches usually give the agent reward bonuses for visiting novel states to encourage exploration. The bonus is proportional to the mismatch between the

predicted and reality, also known as surprise (Schmidhuber, 2010). One kind of predictive model is the dynamics model, wherein the surprise is the error of the models as predicting the next state given the current state and action (Achiam and Sastry, 2017; Stadie et al., 2015). One critical problem of these approaches is the unwanted bias towards transitions where the prediction target is a stochastic function of the inputs, commonly found in partially observable environments. Recent works focus on improving the features of the predictor's input by adopting representation learning mechanisms such as inverse dynamics (Pathak et al., 2017), variational autoencoder, random/pixel features (Burda et al., 2018a), or whitening transform (Ermolov and Sebe, 2020). Although better representations may improve the reward bonus, they cannot completely solve the problem of stochastic dynamics and thus, fail in extreme cases such as the noisy-TV problem (Burda et al., 2018b).

Besides dynamics prediction, several works propose to predict other quantities as functions of the current state by using autoencoder (Nylend, 2017), episodic memory (Savinov et al., 2018), and random network (Burda et al., 2018b). Burda et al. (2018) claimed that using a deterministic random target network is beneficial in overcoming stochasticity issues. Other methods combine this idea with episodic memory and other techniques, achieving good results in large-scale experiments (Badia et al., 2020; 2019). From an information theory perspective, the notation of surprise can be linked to information gain or uncertainty, and predictive models can be treated as parameterized distributions (Achiam and Sastry, 2017; Houthooft et al., 2016; Still and Precup, 2012). Furthermore, to prevent the agent from unpredictable observations, the reward bonus can be measured by the progress of the model's prediction (Achiam and Sastry, 2017; Lopes et al., 2012; Schmidhuber, 1991). However, these methods are complicated and hard to scale, requiring heavy computing. A different angle to handle stochastic observations during exploration is surprsie minimization (Berseth et al., 2020; Rhinehart et al., 2021). In this direction, the agents get bigger rewards for seeing more familiar states. Such a strategy is somewhat opposite to our approach and suitable for unstable environments where the randomness occurs separately from the agents' actions.

These earlier works rely on the principle of using surprise as an incentive for exploration and differ from our principle that utilizes surprise novelty. Also, our work augments these existing works with a surprise memory module and can be used as a generic plug-in improvement for surprise-based models. We note that our memory formulation differs from the memory-based novelty concept using episodic memory (Badia et al., 2019), momentum memory (Fang et al., 2022), or counting (Bellemare et al., 2016; Tang et al., 2017) because our memory operates on the surprise level, not the state level. In our work, exploration is discouraged not only in frequently visited states but also in states whose surprises can be reconstructed using SM. Our work provides a more general and learnable novelty detection mechanism, which is more flexible than the nearest neighbour search or counting lookup table.

## 5 Discussion

This paper presents Surprise Generator-Surprise Memory (SG+SM) framework to compute surprise novelty as an intrinsic motivation for the reinforcement learning agent. Exploring with surprise novelty is beneficial when there are repeated patterns of surprises or random observations. For example, in the Noisy-TV problem, our SG+SM can harness the agent's tendency to visit noisy states such as watching random TV channels while encouraging it to explore rare events with distinctive surprises. We empirically show that our SM can supplement three surprise-based SGs to achieve more rewards in fewer training steps in three grid-world environments. In 3D navigation without external reward, our method significantly outperforms the baselines. On two strong SGs, our SM also achieve superior results in hard-exploration Atari games within 50 million training frames. Even in the long run, our method maintains a clear performance gap from the baselines, as shown in Montezuma Revenge and Frostbite. If we view surprise as the first-order error between the observation and the predicted, surprise novelty–the retrieval error between the surprise and the reconstructed memory, is essentially the second-order error. It would be interesting to investigate the notion of higher-order errors, study their theoretical properties, and utilize them for intrinsic motivation in our future work.

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

## Appendix

### A $\mathcal{W}$ as Associative Memory

This section will connect the associative memory concept to neural networks trained with the reconstruction loss as in Eq. 5. We will show how the neural network ($\mathcal{W}$) stores and retrieves its data. We will use 1-layer feed-forward neural network $W$ to simplify the analysis, but the idea can extend to multi-layer feed-forward neural networks. For simplicity, assuming $W$ is a square matrix, the objective is to minimize the difference between the input and the output of $W$:

For simplicity, assuming $W$ is a square matrix, the objective is to minimize the difference between the input and the output of $W$:

$$\mathcal{L} = \|Wx - x\|_2^2 \tag{6}$$

Using gradient descent, we update $W$ as follow,

$$
\begin{aligned}
W &\leftarrow W - \alpha \frac{\partial \mathcal{L}}{\partial W} \\
&\leftarrow W - 2\alpha \left(Wx - x\right) x^\top \\
&\leftarrow W - 2\alpha W x x^\top + 2\alpha x x^\top \\
&\leftarrow W \left(I - 2\alpha x x^\top\right) + 2\alpha x x^\top
\end{aligned}
$$

where $I$ is the identity matrix, $x$ is the column vector. If a batch of inputs $\{x_i\}_{i=1}^B$ is used in computing the loss in Eq. 6, at step $t$, we update $W$ as follows,

$$W_t = W_{t-1}\left(I - \alpha X_t\right) + \alpha X_t$$

where $X_t = 2\sum_{i=1}^B x_i x_i^\top$. From $t = 0$, after $T$ updates, the weight becomes

$$W_T = W_0 \prod_{t=1}^T \left(I - \alpha X_t\right) - \alpha^2 \sum_{t=2}^T X_t X_{t-1} \prod_{k=t+1}^T \left(I - \alpha X_k\right) + \alpha \sum_{t=1}^T X_t \tag{7}$$

Given the form of $X_t$, $X_t$ is symmetric positive-definite. Also, as $\alpha$ is often very small ($0<\alpha \ll 1$), we can show that $\|I - \alpha X_t\| < 1 - \lambda_{min}\left(\alpha X_t\right) < 1$. This means as $T \to \infty$, $\left\|W_0 \prod_{t=1}^T \left(I - \alpha X_t\right)\right\| \to 0$ and thus, $W_T \to \alpha^2 \sum_{t=2}^T X_t X_{t-1} \prod_{k=t+1}^T \left(I - \alpha X_k\right) + \alpha \sum_{t=1}^T X_t$ independent from the initialization $W_0$. Eq. 7 shows how the data ($X_t$) is integrated into the neural network weight $W_t$. The other components such as $\alpha^2 \sum_{t=2}^T X_t X_{t-1} \prod_{k=t+1}^T \left(I - \alpha X_k\right)$ can be viewed as additional encoding noise. Without these components (by assuming $\alpha$ is small enough),

$$
\begin{aligned}
W_T &\approx \alpha \sum_{t=1}^T X_t \\
&= \alpha \sum_{t=1}^T \sum_{i=1}^B x_{i,t} x_{i,t}^\top
\end{aligned}
$$

or equivalently, we have the Hebbian update rule:

$$W \leftarrow W + x_{i,t} \otimes x_{i,t}$$

where $W$ can be seen as the memory, $\otimes$ is the outer product and $x_{i,t}$ is the data or item stored in the memory. This memory update is the same as that of classical associative memory models such as Hopfield network and Correlation Matrix Memory (CMM) .

Given a query $q$, we retrieve the value in $W$ as output of the neural network:

$$q' = q^\top W$$
$$= q^\top R + \alpha \sum_{t=1}^{T} q X_t$$
$$= q^\top R + 2\alpha \sum_{t=1}^{T} \sum_{i=1}^{B} q^\top x_{i,t} x_{i,t}^\top$$

where $R = W_0 \prod_{t=1}^{T} (I - \alpha X_t) - \alpha^2 \sum_{t=2}^{T} X_t X_{t-1} \prod_{k=t+1}^{T} (I - \alpha X_k)$. If $q$ is presented to the memory $W$ in the past as some $x_j$, $q'$ can be represented as:

$$q' = q^\top R + 2\alpha \sum_{t=1}^{T} \sum_{i=1,i\neq j}^{B} q^\top x_{i,t} x_{i,t}^\top + 2\alpha q^\top \left(qq^\top\right)$$
$$= \underbrace{q^\top R}_{\text{noise}} + \underbrace{2\alpha \sum_{t=1}^{T} \sum_{i=1,i\neq j}^{B} q^\top x_{i,t} x_{i,t}^\top}_{\text{cross talk}} + 2\alpha \|q\| q^\top$$

Assuming that the noise is insignificant thanks to small $\alpha$, we can retrieve exactly $q$ given that all items in the memory are orthogonal[2]. As a result, after scaling $q'$ with $1/2\alpha$, the retrieval error ($\left\| \frac{q'}{2\alpha} - q \right\|$) is 0. If $q$ is new to $W$, the error will depend on whether the items stored in $W$ are close to $q$. Usually, the higher the error, the more novel $q$ is w.r.t $W$.

## B  SM's IMPLEMENTATION DETAIL

In practice, the short-term memory $\mathcal{M}$ is a tensor of shape $[B, N, d]$ where $B$ is the number of actors, $N$ the memory length and $d$ the slot size. $B$ is the SG's hyperparameters and tuned depending on tasks based on SG performance. For example, for the Noisy-TV, we tune RND as the SG, obtaining $B = 64$ and directly using them for $\mathcal{M}$. $N$ and $d$ are the special hyperparameters of our method. As mentioned in Sec. 3.4, we fix $N = 128$ and $d = 16$ in all experiments. As $B$ increases in large-scale experiments, memory storage for $\mathcal{M}$ can be demanding. To overcome this issue, we can use the uniform writing trick to optimally preserve information while reducing $N$ (Le et al., 2019).

Also, for $\mathcal{W}$, by using a small hidden size, we can reduce the requirement for physical memory significantly. Practically, in all experiments, we implement $\mathcal{W}$ as a 2-layer feed-forward neural network with a hidden size of 32 ($2n \to 32 \to 2n$). The activation is tanh. With $n = 512$ $d = 16$, the number of parameters of $\mathcal{W}$ is only about 65K. Also, $Q \in \mathbb{R}^{n \times d}$ and $V \in \mathbb{R}^{d \times n}$ have about 8K parameters. In total, our SM only introduces less than 90K trainable parameters, which are marginal to that of the SG and policy/value networks (up to 10 million parameters).

The join training of SG+SM is presented in Algo. 2. We note that vector notations in the algorithm are row vectors. For simplicity, the algorithm assumes 1 actor. In practice, our algorithm works with multiple actors and mini-batch training.

## C  INTRINSIC REWARD NORMALIZATION

Following (Burda et al., 2018b), to make the intrinsic reward on a consistent scale, we normalized the intrinsic reward by dividing it by a running estimate of the standard deviations

---

[2]By certain transformation, this condition can be reduced to linear independence

---
**Algorithm 1** Intrinsic rewards computing via SG+SM framework.

---
**Require:** $u_t$, and our surprise memory SM consisting of a slot-based memory $\mathcal{M}$, parameters $Q$, $V$, and a neural network $\mathcal{W}$
1: Compute $\mathcal{L}_{SG} = \|u_t\|$
2: Query $\mathcal{M}$ with $u_t$, retrieve $u_t^e = w_t \mathcal{M} V$ where $w_t$ is the attention weight
3: Compute $\mathcal{L}_{\mathcal{M}} = \|u_t^e - u_t.detach()\|$
4: Query $\mathcal{W}$ with $q_t = [u_t^e, u_t]$, retrieve $\tilde{q}_t = \mathcal{W}(q_t)$
5: Compute intrinsic reward $r_t^i = L_{\mathcal{W}} = \|\tilde{q}_t - q_t.detach()\|$
6: **return** $\mathcal{L}_{SG}$, $\mathcal{L}_{\mathcal{M}}$, $L_{\mathcal{W}}$

---

---
**Algorithm 2** Jointly training SG+SM and the policy.

---
**Require:** buffer, policy $\pi_\theta$, surprise-based predictor SG, and our surprise memory SM consisting of a slot-based memory $\mathcal{M}$, parameters $Q$, $V$, and a neural network $\mathcal{W}$
1: Initialize $\pi_\theta$, SG, $Q$, $\mathcal{W}$
2: **for** $iteration = 1, 2, ...$ **do**
3:     **for** $t = 1, 2, ...T$ **do**
4:         Execute policy $\pi_\theta$ to collect $s_t$, $a_t$, $r_t$, forming input $I_t = s_t, ...$ and target $O_t$
5:         Compute surprise $u_t = SG(I_t) - O_t.detach()$ (Eq. 1)
6:         Compute intrinsic reward $r_t^i$ using Algo. 1
7:         Compute final reward $r_t \leftarrow r_t + \beta r_t^i / r_t^{std}$
8:         Add $(I_t, O_t, s_{t-1}, s_t, a_t, r_t)$ to buffer
9:         Add $u_t Q$ to $\mathcal{M}$
10:        **if** done episode **then** clear $\mathcal{M}$
11:     **end for**
12:     **for** $k = 1, 2, .., K$ **do**
13:         Sample $I_t$, $O_t$ from buffer
14:         Compute surprise $u_t = SG(I_t) - O_t.detach()$ (Eq. 1)
15:         Compute $\mathcal{L}_{SG}$, $\mathcal{L}_{\mathcal{M}}$, $L_{\mathcal{W}}$ using Algo. 1
16:         Update SG, $Q$ and $\mathcal{W}$ by minimizing the loss $\mathcal{L} = \mathcal{L}_{SG} + \mathcal{L}_{\mathcal{M}} + \mathcal{L}_{\mathcal{W}}$
17:         Update $\pi_\theta$ with sample $(s_{t-1}, s_t, a_t, r_t)$ from buffer using backbone algorithms
18:     **end for**
19: **end for**

---

of the intrinsic returns. This normalized intrinsic reward (NIR) will be used for training. In addition, there is a hyperparameter named intrinsic reward coefficient to scale the intrinsic contribution relatively to the external reward. We denote the running mean's standard deviations and intrinsic reward coefficient as $r_t^{std}$ and $\beta$, respectively, in Algo. 2. In our experiments, if otherwise stated, $\beta = 1$.

We note that when comparing the intrinsic reward at different states in the same episode (as in the experiment section), we normalize intrinsic rewards by subtracting the mean, followed by a division by the standard deviation of all intrinsic rewards in the episode. Hence, the mean-normalized intrinsic reward (MNIR) in these experiments is different from the one used in training and can be negative. We argue that normalizing with mean and std. of the episode's intrinsic rewards is necessary to make the comparison reasonable. For example, in an episode, method A assigns all steps with intrinsic rewards of 200; and method B assigns novel steps with intrinsic rewards of 1 while others 0. Clearly, method A treats all steps in the episode equal, and thus, it is equivalent to giving no motivation for all of the steps in the episode (the learned policy will not motivate the agent to visit novel states). On the contrary, method B triggers motivation for novel steps in the episodes (the learned policy will encourage visits to novel states). Without normalizing by mean subtraction, it is tempting to conclude that the relative intrinsic reward of method A for a novel step is higher, which is technically incorrect.

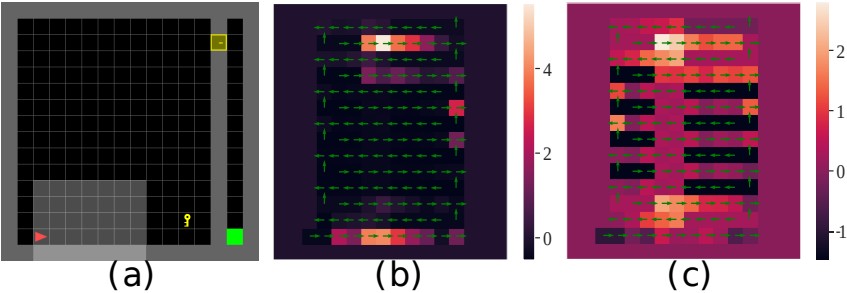

Figure 5: Key-Door: (a) Example map in Key-Door where the light window is the agent's view window (state). MNIR produced for each cell in a manually created trajectory for RND+SM (b) and RND (c). The green arrows denote the agent's direction at each location. The brighter the cell, the higher MNIR assigned to the corresponding state.

## D    EXPERIMENTAL DETAILS

### D.1    NOISY-TV

We create the Noisy-TV environment by modifying the Maze environment (MazeS3Fast-v0) in the MiniWorld library (Apache License) (Chevalier-Boisvert, 2018). The backbone RL algorithm is PPO. We adopt a public code repository for the implementation of PPO and RND (MIT License)[3]. In this environment, the state is an image of the agent's viewport. The details of architecture and hyperparameters of the backbone and RND is presented in Table 4. Most of the setting is the same as in the repository. We only tune the number of actors (32, 128, 1024), mini-batch size (4, 16, 64) and $\epsilon$-clip (0.1, 0.2, 0.3) to suit our hardware and the task. After tuning with RND, we use the same setting for our RND+SM.

Fig. 6 reports all results for this environment. Fig. 6 (a) compares the final intrinsic reward (IR) generated by RND and RND+SM over training time. Overall, RND's IR is always higher than RND+SM's, indicating that our method is significantly reduces the attention of the agent to the noisy TV by assigning less IR to watching TV. Fig. 6 (b) compares the number of noisy actions between two methods where RND+SM consistently shows fewer watching TV actions. That confirms RND+SM agent is less distracted by the TV.

As mentioned in the main text, RND+SM is better at handling noise than RND. Note that RND aims to predict the transformed states by minimizing $\|SG(s_t) - f_R(s_t)\|$ where $f_R$ is a fixed neural network initialized randomly. If RND can learns the transformation, it can pass-through the state, which is similar to reconstruction in an autoencoder. However, learning $f_R$ can be harder and require more samples than learning an identity transformation since $f_R$ is non-linear and complicated. Hence, it may be more challenging for RND to pass-through the noise than SM.

Another possible reason lies in the operating space (state vs. surprise). If we treat white noise as a random variable $X$, a surprise generator (SG) can at most learn to predict the mean of this variable and compute the surprise $U = \mathbb{E}[X|Y] - X$ where $Y$ is a random factor that affects the training of the surprise generator. The factor $Y$ makes the SG produce imperfect reconstruction $\mathbb{E}[X|Y]$[4]. Here, SG and SM learn to reconstruct $X$ and $U$, respectively. We can prove that the variance of each feature dimension in $U$ is smaller than that of $X$ (see Sec. E). Learning an autoencoder on surprise space is more beneficial than in state space since the data has less variance and thus, it may require less data points to learn the data distribution.

Fig. 6 (c) reports performance of all baselines. Besides RND and RND+SM, we also include PPO without intrinsic reward as the vanilla Baseline for reference. In addition, we investigate a simple implementation of SM using count-based method to measure surprise novelty. Concretely, we use SimHash algorithm to count the number of surprise $c(u_t)$ in a similar manner as (Bellemare et al., 2016) and name the baseline RND+SM (count). The

---

[3]https://github.com/jcwleo/random-network-distillation-pytorch
[4]In this case, the perfect reconstruction is $\mathbb{E}[X]$

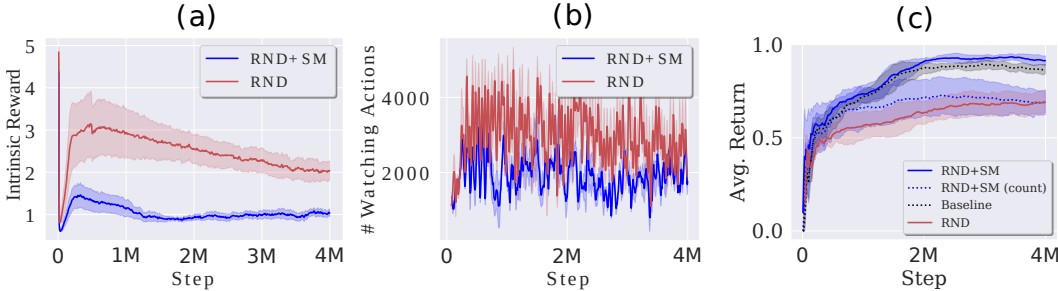

Figure 6: Noisy-TV's learning curves over training steps: (a) Average NIR. (b) Number of watching actions. (c) Average return. All curves are averaged over 5 runs (mean±std).

intrinsic reward is then $\beta/\sqrt{c(u_t)}$. We tune the hyperparameter $\beta = \{0.5, 1, 5\}$ and the hash matrix size $k_h = \{32, 64, 128, 256\}$ and use the same normalization and training process to run this baseline. We report the learning curves of the best variant with $\beta = 0.5$ and $k_h = 128$. The result demonstrates that the proposed SM using memory-augmented neural networks outperforms the count-based SM by a significant margin. One possible reason is that count-based method cannot handle white noise: it always returns high intrinsic rewards. In contrast, our SM can somehow reconstruct white noise via pass-through mechanism and thus reduces the impact of fake surprise on learning. Also, the proposed SM is more flexible than the count-based counterpart since it learns to reconstruct from the data rather than using a fix counting scheme. The result also shows that RND+SM outperforms the vanilla Baseline. Although the improvement is moderate (0.9 vs 0.85), the result is remarkable since the Noisy-TV is designed to fool intrinsic motivation methods and among all, only RND+SM can outperform the vanilla Baseline.

## D.2 MiniGrid

The tasks in this experiment are from the MiniGrid library (Apache License) (Chevalier-Boisvert et al., 2018). In MiniGrid environments, the state is a description vector representing partial observation information such as the location of the agents, objects, moving directions, etc. The three tasks use hardest maps:

- DoorKey: MiniGrid-DoorKey-16x16-v0
- LavaCrossing: MiniGrid-LavaCrossingS11N5-v0
- DynamicObstacles: MiniGrid-Dynamic-Obstacles-16x16-v0

The SGs used in this experiment are RND (Burda et al., 2018b), ICM (Pathak et al., 2017), NGU (Badia et al., 2019) and AE. Below we describe the input-output structure of these SGs.

- RND: $I_t = s_t$ and $O_t = f_R(s_t)$ where $s_t$ is the current state and $f_R$ is a neural network that has a similar structure as the prediction network, yet its parameters are initialized randomly and fixed during training.
- ICM: $I_t = (s_{t-1}, a_t)$ and $O_t = s_t$ where $s$ is the embedding of the state and $a$ the action. We note that in addition to the surprise loss (Eq. 2), ICM is trained with inverse dynamics loss.
- NGU: This agent reuses the RND as the SG ($I_t = s_t$ and $O_t = f_R(s_t)$) and combines the surprise norm with an KNN episodic reward. When applying our SM to NGU, we only take the surprise-based reward as input to the SM. The code for NGU is based on this public repository https://github.com/opendilab/DI-engine.
- AE: $I_t = s_t$ and $O_t = s_t$ where $s$ is the embedding of the state. This SG can be viewed as an associative memory of the observation, aiming to remember the states. This baseline is designed to verify the importance of surprise modeling. Despite sharing a similar architecture, it differs from our SM, which operates on surprise and have an augmented episodic memory to support reconstruction.

| Hyperparameters | ICM | AE |
|---|---|---|
| PPO's state encoder | 3-layer feedforward net (Tanh, h=256) | 3-layer feedforward net (Tanh, h=256) |
| SG's surprise predictor | 4-layer feedforward net (ReLU, h=512) | 3-layer feedforward net (Tanh, h=512) |
| Intrisic Coef. $\beta$ | 1 | 1 |
| Num. Actor $B$ | 64 | 64 |
| Minibatch size | 64 | 64 |
| Horizon $T$ | 128 | 128 |
| Adam Optimizer's lr | $10^{-4}$ | $10^{-4}$ |
| Discount $\gamma$ | 0.999 | 0.999 |
| Intrinsic $\gamma^i$ | 0.99 | 0.99 |
| GAE $\lambda$ | 0.95 | 0.95 |
| PPO's clip $\epsilon$ | 0.2 | 0.2 |

Table 3: Hyperparameters of ICM and AE (PPO backbone).

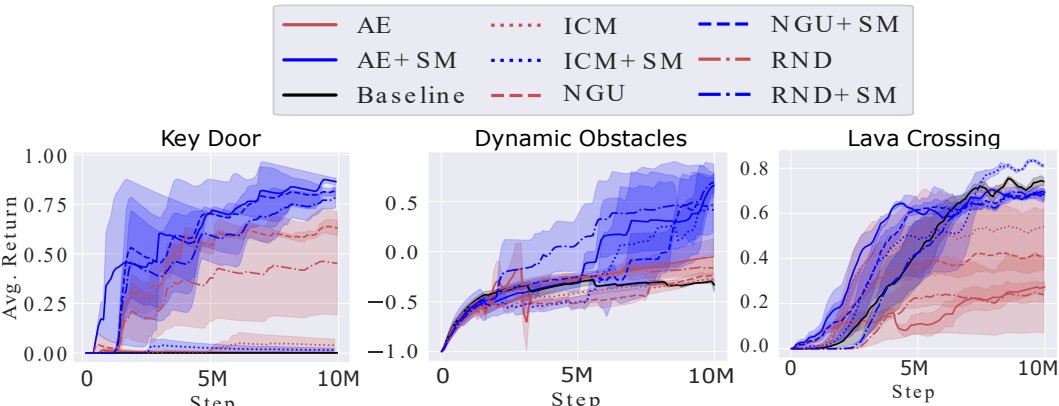

Figure 7: Minigrid's learning curves over 10 million training steps (mean±std. over 5 runs).

The backbone RL algorithm is PPO. The code for PPO and RND is the same as in Sec. D.1. We adopt a public code repository for the implementation of ICM (MIT License)[5]. We implement AE ourselves using a 3-layer feed-forward neural network. For the SGs, we only tune the number of actors (32, 128, 1024), mini-batch size (4, 16, 64) and $\epsilon$-clip (0.1, 0.2, 0.3) for the DoorKey task. We also tune the architecture of the AE (number of layers: 1,2 or 3, activation tanh or ReLU) on the DoorKey task. After tuning the SGs, we use the same setting for our SG+SM. The detailed configurations of the SGs for this experiment are reported in Table 3 and Table 4.

The full learning curves of the backbone (Baseline), SG and SG+SM are given in Fig. 7. To visualize the difference between surprise and surprise residual vectors, we map these in the synthetic trajectory to 2-dimensional space using t-SNE projection in Fig. 8. The surprise points show clustered patterns for high-MNIR states, which confirms our hypothesis that there exist familiar surprises (they are highly surprising due to high norm, yet repeated). In contrast, the surprise residual estimated by the SM has no high-MNIR clusters. The SM transforms clustered surprises to scatter surprise residuals, resulting in a broader range of MNIR, thus showing significant discrimination on states that have similar surprise norm.

## D.3 ATARI

The Atari 2600 Games task involves training an agent to achieve high game scores. The state is a 2d image representing the screen of the game.

---

[5] https://github.com/jcwleo/curiosity-driven-exploration-pytorch

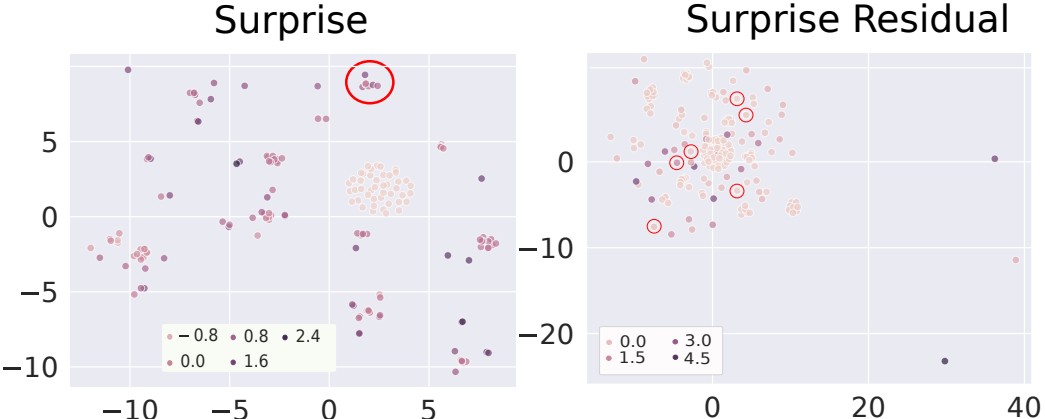

Figure 8: Key-Door: t-NSE 2d representations of surprise $(u_t)$ and surprise residual $(\hat{q}_t - q_t)$. Each point corresponds to the MNIR at some step in the episode. Color denotes the MNIR value (darker means higher MNIR). The red circle on the left picture shows an example cluster of 6 surprise points. Surprise residuals of these points are not clustered, as shown in 6 red circles on the right pictures. In other words, surprise residual can discriminate surprises with similar norms.

| Hyperparameters | MiniGrid | Noisy-TV+MiniWorld | Atari |
|---|---|---|---|
| PPO's state encoder | 3-layer feedforward net (Tanh, h=256) | 3-layer Leaky-ReLU CNN with kernels $\{12/32/8/4, 32/64/4/2, 64/64/3/1\}$ +2-layer feedforward net (ReLU, h=256) | 3-layer Leaky-ReLU CNN with kernels $\{4/32/8/4, 32/64/4/2, 64/64/3/1\}$ +2-layer feedforward net (ReLU, h=256) |
| RND's surprise predictor | 3-layer feedforward net (Tanh, h=512) | 3-layer Leaky-ReLU CNN with kernels $\{1/32/8/4, 32/64/4/2, 64/64/3/1\}$ +2-layer feedforward net (ReLU, h=512) | 3-layer Leaky-ReLU CNN with kernels $\{1/32/8/4, 32/64/4/2, 64/64/3/1\}$ +2-layer feedforward net (ReLU, h=512) |
| Intrinsic Coef. $\beta$ | 1 | 1 | 1 |
| Num. Actor $B$ | 64 | 64 | 128 |
| Minibatch size | 64 | 64 | 4 |
| Horizon $T$ | 128 | 128 | 128 |
| Adam Optimizer's lr | $10^{-4}$ | $10^{-4}$ | $10^{-4}$ |
| Discount $\gamma$ | 0.999 | 0.999 | 0.999 |
| Intrinsic $\gamma^i$ | 0.99 | 0.99 | 0.99 |
| GAE $\lambda$ | 0.95 | 0.95 | 0.95 |
| PPO's clip $\epsilon$ | 0.2 | 0.1 | 0.1 |

Table 4: Hyperparameters of RND (PPO backbone).

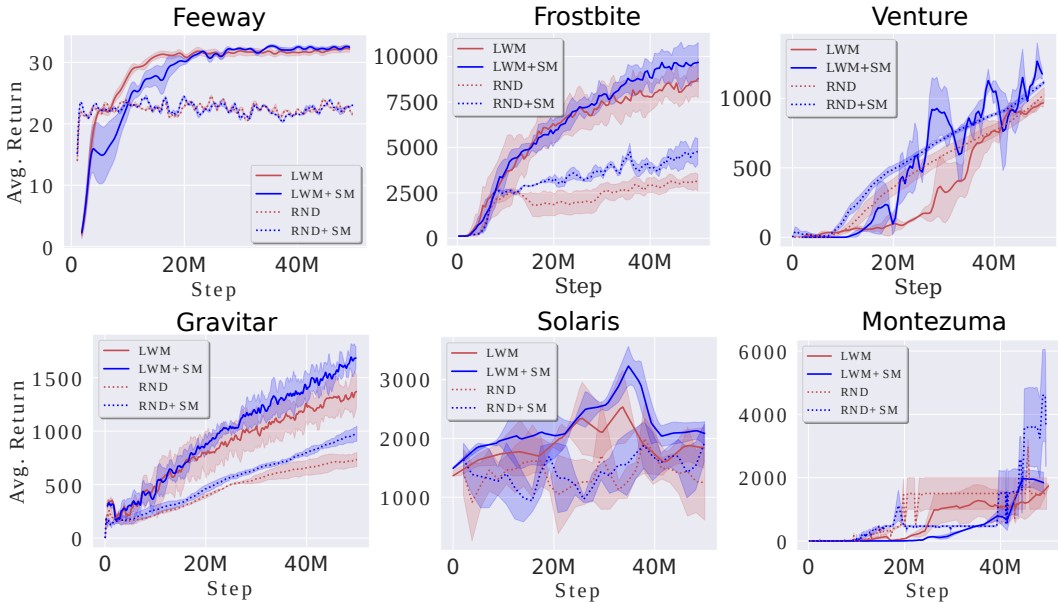

Figure 9: Atari low-sample regime: learning curves over 50 million frames (mean±std. over 5 runs). To aid visualization, we smooth the curves by taking average over a window sized 50.

**SG and RL backbone implementations**    We use 2 SGs: RND and LWM. RND uses a PPO backbone as in previous sections. On the other hand, LWM uses DQN backbone with CNN-based encoder and GRU-based value function. The LWM SG uses GRU to model forward dynamics of the environment and thus its input is: $I_t = (s_{t-1}, a_t, h_{t-1})$ where $s_{t-1}$ is the embedding of the previous state, $a_t$ the current action, and $h_{t-1}$ the hidden state of the world model GRU. The target $O_t$ is the embedding of the current state $s_t$.

RND follows the same implementation as in previous experiments. We use the public code of LWM provided by the authors[6] to implement LWM. The hyperparameters of RND and LWM are tuned by the repository's owner (see Table 4 for RND and refer to the code or the original paper (Ermolov and Sebe, 2020) for the details of LWM implementation). We augment them with our SM of default hyperparameters $N = 128, d = 16$.

**Training and evaluation**    We follow the standard training for Atari games, such as stacking four frames and enabling sticky actions. All the environments are based on OpenAI's gym-atari's NoFrameskip-v4 variants (MIT Liscence)[7] . After training, we evaluate the models by measuring the average return over 128 episodes and report the results in Table. 2. Depending on the setting, the models are trained for 50 or 200 million frames.

**Results**    Fig. 9 demonstrates the learning curves of all models in 6 Atari games under the low-sample regime. LWM+SM and RND+SM clearly outperfrom LWM and RND in Frost-bite, Venture, Gravitar, Solaris and Frostbite, Venture, Gravitar and MontezumaRevenge, respectively. Table 5 reports the results of more baselines.

## D.4    ABLATION STUDY

**Role of Memories**    We conduct more ablation studies to verify the need for the short $\mathcal{M}$ and long-term ($\mathcal{W}$) memory in our SM. We design additional baselines SM (no $\mathcal{W}$) and SM (no $\mathcal{M}$) (see Sec. 3.4), and compare them with the SM with full features in Montezuma Revenge and Frostbite task. Fig. 10 (a) shows that only SM (full) can reach an average score of more than 5000 after 50 million training frames. Other ablated baselines can only achieve around 2000 scores.

---

[6]https://github.com/htdt/lwm
[7]https://github.com/openai/gym

| Task | EMI♠ | EX2♠ | ICM♠ | AE-SH♠ | LWM♠ | RND♠ | LWM◇ | LWM+SM◇ | RND◇ | RND+SM◇ |
|---|---|---|---|---|---|---|---|---|---|---|
| Freeway | **33.8** | 27.1 | 33.6 | 33.5 | 30.8 | 33.3 | 31.1 | 31.6 | 22.2 | 22.2 |
| Frostbite | 7,002 | 3,387 | 4,465 | 5,214 | 8,409 | 2,227 | 8,598 | ***10,258*** | 2,628 | *5,073* |
| Venture | 646 | 589 | 418 | 445 | 998 | 707 | 985 | ***1,381*** | 1,081 | *1,119* |
| Gravitar | 558 | 550 | 424 | 482 | 1,376 | 546 | 1,242 | ***1,693*** | 739 | *987* |
| Solaris | 2,688 | 2,276 | 2,453 | **4,467** | 1,268 | 2,051 | 1,839 | *2,065* | 2,206 | *2,420* |
| Montezuma | 387 | 0 | 161 | 75 | 2,276 | 377 | 2,192 | 2,269 | 2,475 | **5,187** |
| Norm. Mean | 61.4 | 40.5 | 46.1 | 52.4 | 80.6 | 42.2 | 80.5 | ***97.0*** | 50.7 | *74.8* |
| Norm. Median | 34.9 | 32.3 | 23.1 | 33.3 | 60.8 | 32.7 | 66.5 | *83.7* | 58.3 | ***84.6*** |

Table 5: Atari: test performance after 50 million training frames (mean over 5 runs). ♠ is from a prior work (Ermolov and Sebe, 2020). ◇ is our run. The last two rows are mean and median human normalized scores. Bold denotes best results. Italic denotes that SG+SM is significantly better than SG regarding Cohen effect size less than 0.5.

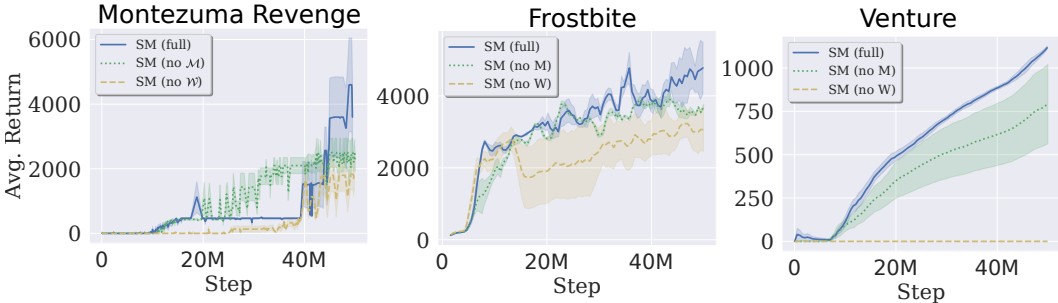

Figure 10: Ablation study: average returns (mean±std.) over 5 runs.

We also shows the impact of the episodic memory in decreasing the intrinsic rewards for similar states as discussed in Sec. 2.3. We select 3 states in the MiniGrid's KeyDoor task and computes the MNIR for each state, visualized in Fig. 11. At the step-1 state, the MNIR is low since there is nothing special in the view of the agent. At the step-15 state, the agent first sees the key, and get a high MNIR. At the step-28 state, the agent drops the key and sees the key again. This event is still more interesting than the step-1 state. However, the view is similar to the one in step 15, and thus, the MNIR decreases from 0.7 to 0.35 as expected.

**No Task Reward**   The tasks in this experiment are from the MiniWorld library (Apache License) (Chevalier-Boisvert, 2018). The two tasks are:

- Easy: MiniWorld-PickupObjs-v0
- Hard: MiniWorld-FourRooms-v0

The backbone and SG are the same as in Sec. D.1. We remove the task/external reward in this experiment. For the Baseline, without task reward, it receives no training signal

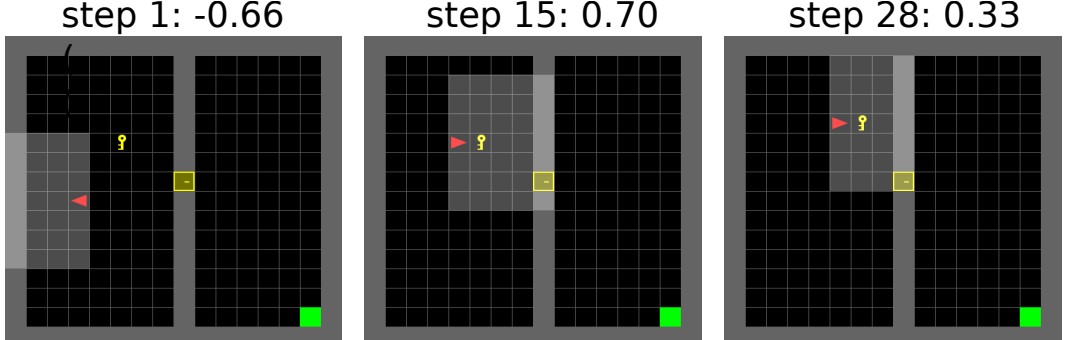

Figure 11: MiniGrid's KeyDoor: MNIR of SM at different steps in an episode.

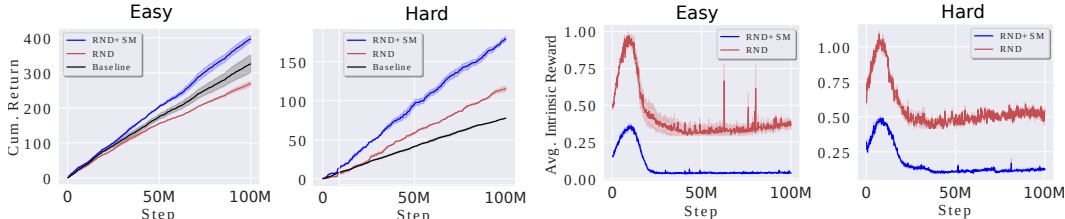

Figure 12: MiniWorld: Exploration without task reward. Left: Cumulative task returns over 100 million training steps for two setting: Easy (1 room, 3 objects) and Hard (4 rooms, 1 object). Right: The average intrinsic reward over training time. The learning curves are taken average (mean±std.) over 5 runs

and thus, showing a similar behavior as a random agent. Fig. 12 illustrates the running average of cumulative task return and the intrinsic reward over training steps. In the Easy mode, the random Baseline can even perform better than RND, which indicates that biased intrinsic reward is not always helpful. RND+SM, in both modes, shows superior performance, confirming that its intrinsic reward is better to guide the exploration than that of RND.

## E    THEORETICAL PROPERTY OF SURPRISE SPACE'S VARIANCE

Let $X$ be a random variable representing the observation at some timestep, a surprise generator (SG) can at most learn to predict the mean of this variable and compute the surprise $U = \mathbb{E}[X|Y] - X$ where $Y$ is a random factor that affect the prediction of SG and makes it produce imperfect reconstruction $\mathbb{E}[X|Y]$ instead of $\mathbb{E}[X]$. For instance, in the case of an autoencoder AE as the SG, $X$ and $U$ are $s_t$ and $AE(s_t) - s_t$, respectively.

Let us denote $Z = \mathbb{E}(X|Y)$, then $\mathbb{E}[Z|Y] = Z$ and $\mathbb{E}[Z^2|Y] = Z^2$. We have

$$\begin{aligned}
\text{var}(X) &= \text{var}(X - Z + Z) \\
&= \text{var}(X - Z) + \text{var}(Z) + 2\text{cov}(X - Z, Z) \\
&= \text{var}(X - Z) + \text{var}(Z) + 2\mathbb{E}[(X - Z)Z] - 2\mathbb{E}[X - Z]\mathbb{E}[Z]
\end{aligned}$$

Using the Law of Iterated Expectations, we have

$$\begin{aligned}
\mathbb{E}[X - Z] &= \mathbb{E}[\mathbb{E}[X - Z|Y]] \\
&= \mathbb{E}[\mathbb{E}[X|Y] - \mathbb{E}[Z|Y]] \\
&= \mathbb{E}[Z - Z] = 0
\end{aligned}$$

and

$$\begin{aligned}
\mathbb{E}[(X - Z)Z] &= \mathbb{E}[\mathbb{E}[(X - Z)Z|Y]] \\
&= \mathbb{E}[\mathbb{E}[XZ - Z^2|Y]] \\
&= \mathbb{E}[\mathbb{E}(XZ|Y) - \mathbb{E}(Z^2|Y)] \\
&= \mathbb{E}[Z\mathbb{E}(X|Y) - Z^2] \\
&= \mathbb{E}[Z^2 - Z^2] = 0
\end{aligned}$$

Therefore,

$$\text{var}(X) = \text{var}(X - Z) + \text{var}(Z)$$

Let $C_{ii}^X$, $C_{ii}^{X-Z}$ and $C_{ii}^Z$ denote the diagonal entries of these covariance matrices, they are the variances of the components of the random vector $X$, $X - Z$ and $Z$, respectively. That is,

$$\left(\sigma_i^X\right)^2 = \left(\sigma_i^{X-Z}\right)^2 + \left(\sigma_i^Z\right)^2$$
$$\Rightarrow \left(\sigma_i^X\right)^2 \geq \left(\sigma_i^{X-Z}\right)^2 = \left(\sigma_i^U\right)^2$$

In our setting, $X$ and $U$ represents observation and surprise spaces, respectively. Therefore, the variance of each feature dimension in surprise space is smaller than that of observation space. The equality is obtained when $\left(\sigma_i^Z\right)^2 = 0$ or $\mathbb{E}\left(X|Y\right) = \mathbb{E}\left(X\right)$. That is, the SG's prediction is perfect, which is unlikely to happen in practice.

## F   LIMITATIONS

Our method assumes that surprises have patterns and can be remembered by our surprise memory. There might exist environments beyond those studied in this paper where this assumption may not hold, or surprise-based counterparts already achieve optimal exploration (e.g., perfect SG) and thus do not need SM for improvement (e.g., Freeway game). In addition, $\mathcal{M}$ and $\mathcal{W}$ require additional physical memory (RAM/GPU) than SG methods. Finally, a plug-in module like SM introduces more hyperparameters, such as N and d. Although we find the default values of N=128 and d=16 work well across all experiments in this paper, we recommend adjustments if users apply our method to novel domains.

