# OpenReview forum: "Intrinsic Motivation via Surprise Memory"
_ICLR.cc/2023/Conference — Submitted to ICLR 2023_

### Official Review · Reviewer_oPBQ · 2022-10-16

**Confidence:** 4
**Correctness:** 3
**Technical Novelty And Significance:** 3
**Empirical Novelty And Significance:** 3
**Recommendation:** 8

**Clarity, Quality, Novelty And Reproducibility:**

The work is easy to follow and clearly written. It could be fixed following the suggestions I wrote above but it can be easily fixed. The only important thing is that authors didn't share the code neither could I find any mentions to it in the appendix. Would be a good thing to have for reproducibility.

I couldn't find any flaws in the proofs at the appendix about their theoretical propositions. Also, they provide abundant detail on hyperparameters, pseudocode and experiment details there.

The paper has abundant experiments, ablations, baselines and benchmarks and the results are quite promising. As far as I know the method itself is novel, I only know of [1] that also works with intrinsic motivation and memory. The method presented here is quite different from that one, but probably it is worth to add to the related work.


[1]Fang, Zheng, Biao Zhao, and Guizhong Liu. "Image Augmentation Based Momentum Memory Intrinsic Reward for Sparse Reward Visual Scenes." arXiv preprint arXiv:2205.09448 (2022).

**Strength And Weaknesses:**

I believe that this is a good work, the method is novel and non-trivial, authors provide abundant theoretical and experimental ground for their approach. The paper is also well presented and easy to follow.

Weaknesses:
1) The authors only present the strengths of this method, which seem to be many, but I could not see any discussion on limitations or main weaknesses of this approach. The future lines proposed at the end feel a little hand waving in my opinion, it doesn't provide much ground for future researchers to continue this work.

2) The need of the AE, while it is clearly demonstrated in the ablation experiments, it is lacking an explanation of why in the theoretical side -besides that it works better-. Going through the work I could grasp some intuition, but would be good that the authors detail this further in section 2 since intuitively, one could think that the norm between the surprise of this state and the one in memory would already tell you if this is an "expected" surprise.

Extra:
I don't want to point this as a weakness but I would like to ask the authors about how does the SM intrinsic reward evolves through training when you are training for the same objective for long. What I mean is that, if you are training your agent to get a red box at the end of the episode, since the box is exotic it provides high intrinsic reward. However, as you train and train the agent the AE will get better to recover that red box surprise at the end, contrarily, the AE probably cannot learn so well to reconstruct the tv noise. My question is, does the intrinsic reward for that final goal get relatively lower to the one of the tv as training progresses? And if not, could Authors explain why it doesn't happen?


**Summary Of The Paper:**

This work introduces a new method for intrinsic motivation in RL called Surprise Memory (SM), that in similar fashion to surprise-based intrinsic motivation, provides an intrinsic reward to the agent when it finds a 'new' observation. SM differs in that it takes in account if this 'surprise' was already expected from previous memories and reduce the intrinsic reward if that is the case.

Intuitively, the method consists of a memory module (of fixed size) with previous surprises and an autoencoder (AE) that tries to reconstruct a query with the current surprise and the closest matching surprise within memory. The worse the AE does, the larger the intrinsic reward.

**Summary Of The Review:**

Good paper, novel method, targets one of the main issues of intrinsic motivation with promising results. The work seems solid and sound although could be improved by detailing weaknesses and limitations and with further motivation of the AE mechanism.

---

> ### Author Response · Authors · 2022-11-16
> **Reply to Reviewer oPBQ**
>
> Thank you for your positive comments. We address your concerns below.
>
> > "Limitation"
>
> Thank you for your suggestion. We have added the following description of our method's limitations in Appendix F and will move to the main text if more space is allowed.
>
> Our method assumes that surprises have patterns and can be remembered by our surprise memory. There might exist environments beyond those studied in this paper where this assumption may not hold, or surprise-based counterparts already achieve optimal exploration (e.g., perfect SG) and thus do not need SM for improvement (e.g., Freeway game). In addition,  M  and  W require additional physical memory (RAM/GPU) than SG methods. Finally, a plug-in module like SM introduces more hyperparameters, such as  N  and d. Although we find the default values of  N=128  and  d=16  work well across all experiments in this paper, we recommend adjustments if users apply our method to novel domains.
>
> > "The need of the AE"
>
> AE is like a long-term inter-episode associative memory. Unlike slot-based memories, AE has a fixed memory capacity, compresses information and learns data representations. We could store the surprise in a slot-based memory across episodes, but the size of this memory would be autonomous, and the data would be stored redundantly. Hence, the quality of the stored surprise will reduce as more and more observations come in. On the other hand, AE can efficiently compress surprises to latent representations and hold them to its neural weights, and the surprise retrieval is optimized. Besides, AE can learn to switch between 2 mechanisms: pass-through and pattern retrieval, to optimally achieve its objective. We cannot do that with slot-based memory. We have added the explanation to the revision.
>
> > "Extra:"
>
> Thank you for your interesting question. There should be a balance point for the reconstructing ability of the AE as long as it keeps minimizing the objective in Eq. 5. To be specific, as the agent sees the red box more frequently, it is true that AE tends to yield a relatively lower intrinsic reward (IR) for the event, leading to nonoptimal policy. But it only lasts for a short time. Since the nonoptimal policy will make the agent visits non-goal states such as watching TV, the AE will be busy learning to reconstruct these unimportant events. Due to limited capacity, the AE will soon forget how to recall the red box and get better at reconstructing other things like walls or passing through noises. Hence, once again, the IR for seeing the red box will be high, aligning the agent to the optimal policy.
>
> > "Clarity, Quality, Novelty And Reproducibility"
>
> Thank you for your suggestion regarding the code. We plan to release our complete code and experiment upon publication. Thank you for pointing out a recent relevant work by Fang et al. (2022). We have included it in the related work.

---

> > ### Comment · Reviewer_oPBQ · 2022-11-20
> > **Response to authors**
> >
> > I want to thank the authors for their detailed response.
> >
> > I believe that the update version makes the work easier to understand and follow. After reading the different reviews and updates I am still maintaining my score as I see this as a fair and novel contribution -existing methods using memory are quite different from the one presented here-, well presented and with good theoretical and empirical coverage.
> >
> > As a side note I think the response to my extra question would be a nice addition in the paper, as this could be a potential line to follow by future iterations of this method -since space is limited I would suggest to explain this in the appendix and give a reference in the conclusions-.

---

### Official Review · Reviewer_ZXKP · 2022-10-24

**Confidence:** 4
**Correctness:** 2
**Technical Novelty And Significance:** 2
**Empirical Novelty And Significance:** 2
**Recommendation:** 3

**Clarity, Quality, Novelty And Reproducibility:**


- The introduction for this paper is less than simple to understand. there is some immediate discussion between surprise and surprise novelty, yet neither of these concepts are explained in sufficient detail within the introduction. The introduction then goes on to somewhat discuss the importance of considering these two different aspects and why memory is helpful but not why memory is necessary.
- More information is needed to see why the diagram in figure 2 is an ideal design to be able to memorize and recover surprising novelty. The paper should also cite related work on being able to memorize normal trajectories in some type of sequence.
- For example, it was not clear until the middle of the second page that surprised novelty might be the second-order version of surprise.
- It's not obvious if proposition one holds. We don't have much information about what $$X$$ and $$U$$ should be.
- The surprise norm appears to be similar to the prediction mechanism from RND. The RND network is learning to estimate the surprise expectation and then comparing that to the current observation. More details that describe the difference between RND and the prediction comparison mechanism used in this work are needed to understand the novelty.
- The details around the discussion of how the episodic memory is used and trained are difficult to understand. Why is there an additional model for training to predict some of the memory while also that a buffer of the most recent history of the agent is kept in order to be able to compute the surprise novelty? A diagram to further explain the process and necessary components to be able to predict or produce the surprise novelty would be very helpful to the reader.
- The writing in section 2 is also challenging to understand which pieces are the novel pieces being introduced in this work? The authors should consider editing a background or prior methods section. An example to include in this section is the autoencoder training section. Training autoencoders is not new and has been around for many years.
- The comparison on the paper should select more environments that are used in prior (ICM and RND) papers.
- Table 2, and in general for the results of the paper, needs to include the confidence information and statistics over this analysis. How many random seeds are used to run this analysis? How much does the distribution of training in these different methods overlap? Preferably a t-test should be performed over the different methods in the paper so we can understand how statistically confident we can be in these results.
- In addition, the results in figure 5 do not appear to show converged policies. It is possible that prior methods end up outperforming the method proposed in this paper if more training time is given. This makes the results shown in figure 5 difficult to use in the assessment.
- Given that a large motivation for this paper is to deal with undesired or unusual stochastic elements of the environment, this paper should also cite the new line of work on surprise minimization[1,2] that, by default, has a well-conceptualized solution to this problem.


**Strength And Weaknesses:**

pros
- The method in the paper does make some analysis in order to be able to correct and adjust some of the challenges with applying curiosity-based exploration bonuses in different types of environments.
- An adequate amount of experimental analysis at the end of the paper indicates the potential for such a method to support a contribution over prior curiosity base metrics.

cons
- Generally, some of the conceptual parts and description of the method are challenging to follow. In particular, some of the description around surprise and surprise novelty and what is the mathematical difference between these concepts is not thoroughly explained. This makes it challenging to understand the paper's novelty and reuse the findings of the paper and future work by the community
. There is additional related work that this paper should cite and compare. In particular, if the paper is citing the noisy TV problem and the general aspect that environments can have stochastic elements in them, then the paper should consider comparing to additional methods that use surprise minimization[1,2].

- [1] Berseth, G., Geng, D., Devin, C. M., Rhinehart, N., Finn, C., Jayaraman, D., & Levine, S. (2021). SMiRL: Surprise Minimizing Reinforcement Learning in Unstable Environments. International Conference on Learning Representations
- [2] Rhinehart, N., Wang, J., Berseth, G., Co-Reyes, J. D., Hafner, D., Finn, C., & Levine, S. (2021). Intrinsic Control of Variational Beliefs in Dynamic Partially-Observed Visual Environments. Advances in Neural Information Processing Systems, 34.

**Summary Of The Paper:**

This method proposes some modifications to curiosity-based exploration methods to help combat some of the corner cases that they don't deal well with, which also results in increased performance. The motivation for this work is that exploration is still a really large challenge and reinforcement learning and even some of the best-proposed methods have some challenges in being able to explore properly in all types of environments (MDPs). The method proposed in this paper introduces a new metric related to surprise novelty, and the surprise novelty helps calculate and reduce the noisy TV problem for curiosity-based metrics. The paper does introduce some interesting ideas, and there's some evaluation at the end of the paper to show that this surprise novelty metric is less susceptible to some of the issues with normal curiosity base methods.

**Summary Of The Review:**

The proposed method for performing better curiosity-based exploration is promising. However, the writing and analysis make it difficult to understand how the method works and if it is novel. These issues will need to be addressed for me to raise my score.

---

> ### Author Response · Authors · 2022-11-16
> **Reply to Reviewer ZXKP (part 1)**
>
> Thank you for your detailed review. We address your concerns in the following.
>
> >"In particular, some of the description around surprise and surprise novelty and what is the mathematical difference between these concepts is not thoroughly explained"
>
> We refer the reviewer to the parts of our original paper explaining the concepts:
>
> - Surprise is defined early in the second paragraph of the Introduction. It is the difference between observation and prediction.
> - Surprise norm is defined in Eq. 1. It is the norm of the surprise vector.
> - Surprise novelty is defined in Eq. 4. It is the norm of the difference between the surprise and reconstructed surprise.
> - The theoretical property of the difference between surprise and surprise norm is given in Sec. 2.1.
>
> > "There is additional related work that this paper should cite and compare"
>
> Thank you for pointing out these relevant works. Surprise minimization encourages agents to receive larger rewards for experiencing more familiar states. Such a strategy is somewhat opposite to our approach and only suitable for unstable environments. We note that Noisy-TV is not in the class of unstable environments because the former triggers randomness when the agent takes a specific action (watch TV). At the same time, in the latter, unexpected events happen without the goal-driven behaviour of the agent. We have included the comparison in the related work section of our revised manuscript.
>
> > "The introduction for this paper  ..."
>
> We use the second and third paragraphs of the Introduction to explain surprise and surprise novelty, respectively. The necessity of memory was explained in the first three sentences of the third paragraph: to give the agent the ability to compare the current surprise with surprises in past encounters.
>
> > "More information is needed to see why the diagram in figure 2 is an ideal design to be able to memorize and recover surprising novelty."
>
> Thank you for your suggestion. Our design effectively recovers surprise novelty because it can handle intra and inter-episode surprise patterns thanks to M and W, respectively. M can quickly adapt and recall surprises that occur within an episode. W is slower and focuses more on consistent surprise patterns across episodes throughout training. We have added the information in the revision.
>
> >"The paper should also cite related work on being able to memorize normal trajectories in some type of sequence."
>
> Thank you for your comment. We note that we do not use memory to store trajectories. What is stored is just a vector representing surprise at a single timestep. We have cited other works using memory to keep states or trajectories for intrinsic motivation, which include NeverGiveUp and Episodic Curiosity.
>
> >"For example, it was not clear until the middle of the second page that surprised novelty might be the second-order version of surprise."
>
> We note that the "second-order version of surprise" is just a different interpretation of the surprise novelty. It is not the definition. The definition is given early in the last sentence of the third paragraph of the Introduction: "The error between the query and the autoencoder's output is defined as surprise novelty.". Please mind that surprise novelty is a new concept defined by a memory retrieval process. The "second-order version of surprise" is just a property.
>
> > "It's not obvious if proposition one holds. "
>
> The proof is given in Appendix E. The only assumption is SG is imperfect. In other words, it cannot predict the observation exactly, which usually happens in practice.
>
> > "We don't have much information about what X and U should be"
>
> X and U are the observation and surprise vectors, repsectively. We have added examples in Appendix E. For instance, in the case of an autoencoder AE as the SG, X and U are $s_t$ and AE($s_t$)-$s_t$, respectively.
>
> > "The surprise norm appears to be similar to the prediction mechanism from RND"
>
> It is the same. There is no difference in the case SG is RND. We want to clarify that surprise norm is not our contribution. It is an old technique used by RND, ICM and other works. We give it a name to differentiate it from our proposal: surprise novelty.
>
> > "The details around the discussion ..."
>
> Due to the space limit, the training and process details (algorithms) are moved to Appendix B for readers to follow. We mentioned that at the end of Sec. 2.
>
> > "Why is there an additional model ..."
>
> We believe the explanation was already given in the Introduction's third paragraph ("A further mechanism ..." ) and Sec. 2.3's second paragraph.
>
> > "Training autoencoders is not new and has been around for many years"
>
> We agree with the Reviewer. That's why we did not describe it in detail. Instead, we only presented the final loss in Eq. (5), focusing on explaining the input/output of the autoencoder. It is the input and output parts are novel since it is the first time an autoencoder is used to store and retrieve surprise.

---

> > ### Author Response · Authors · 2022-11-16
> > **Reply to Reviewer ZXKP (part 2)**
> >
> > > "The comparison on the paper should select more environments that are used in prior (ICM and RND) papers."
> >
> > We chose Atari games (e.g. Montezuma Revenge) as the most common environment used by many prior works. The only difference is that previous works train RL agents in these environments for billion of frames. We, instead, are more interested in the low-sample setting and only compete with other SOTA methods under this regime.
> >
> > > "Table 2 "
> >
> > As mentioned in the caption, we reported the mean over five runs. To save space, we did not present confidence or std, yet the readers can check the full learning curves with mean and std. in the Appendix's Fig. 9. Although we did not run a t-test, we used a simpler Cohen effect size to ensure the gap is significant (see caption).
> >
> > > "Figure 5"
> >
> >  Please note that our focus is sample efficiency, reflecting real-world problems' constraints. In Atari, we show 200 million frame results to show that our method consistently outperforms other baselines in 50M and 200M regimes with a significant margin. This potentially indicates that if trained more, our approach will continue outperforming (In RND paper, after 1.97 billion frames of training, RND only reaches maximum of ~11K score in Montenzuma Revenge, which is not far from our model at 200M frames).
> >
> > > "Given that a large motivation ..."
> >
> > We agree that these works are relevant and have cited them in the revision. However, we note that they are designed for unstable environments, and the authors proposed specific problems satisfying this property. Generally, the environment must be uniquely stochastic to make it reasonable to pursue surprise minimization, which is not the case for our problems.

---

### Official Review · Reviewer_422A · 2022-10-25

**Confidence:** 3
**Correctness:** 4
**Technical Novelty And Significance:** 3
**Empirical Novelty And Significance:** 3
**Recommendation:** 5

**Clarity, Quality, Novelty And Reproducibility:**

The paper is clear, although I believe a better definition of the learning problem onto which the MANN approach is then plugged in would help also non purely RL researchers in benefitting from this work.

Novelty is slightly limited (see above)

The presentation is clear enough for a field expert to reproduce the work.

**Strength And Weaknesses:**

Strengths
- The use of memory on the surprise level is novel and well motivated
- The approach can be integrated easily on existing methods

Weaknesses

- Novelty is slightly limited, memory has been used here https://arxiv.org/pdf/2002.06038.pdf as also mentioned in the paper but with a different data to be stored (agent state vs surprise)
- Results show the improvement with respect to three methods but there are no direct comparison with existing sota. Specifically whye the only other memory based approach (https://arxiv.org/pdf/2002.06038.pdf) is not taken into account in the comparison?
In general a table with recent sota methods reported would improve the overall presentation.


**Summary Of The Paper:**

This paper proposes a new method to manage surprise signals in reinforcement learning. Surprises are stored in a memory for each episode and the memory is wiped after each episode. An autoencoder is used to perform readouts. The memory module can be plugged in existing surprise generators. Benchmark results show that the addition of the memory is almost always beneficial. The main difference with respect to competing memory based approach in RL is that the proposed memory works at the surprise level and not at the state level.



**Summary Of The Review:**

The approach proposed plugs a memory module into RL system using surprise generation. The use of such module is shown to empirically improve results. Novelty is slightly limited and some comparisons are missing.

---

> ### Author Response · Authors · 2022-11-16
> **Reply to Reviewer 422A**
>
> Thank you for your constructive feedback. We will answer your questions below.
> > "Novelty is slightly limited"
>
> Although memory has been used here and there, it is the first time memory is used to compute surprise novelty. Remarkably, the concept of surprise novelty constitutes the novelty of our paper. In addition, our proposed memory is different from the mentioned NGU agent. We do not use KNN to compute the intrinsic reward. Our memory is fully differentiable and learned via minimizing the recall error.
>
> > "There are no direct comparison with existing sota ...  In general a table with recent sota methods reported would improve the overall presentation."
>
> Thank you for your suggestion. Table 2 (or full Table 5) already included the SOTA in Atari low-sample setting (50 million frames). Methods like NGU are trained for much longer (1e10 frames), so they are not included in this table. Although it is not fair to compare, take Montezuma's revenge as an example; looking at the learning curve from the NGU paper, we believe our method outperforms NGU at 50 million frames.
>
> We note that NGU is another SG and can be coupled with our SM. As NGU is slow and our compting resource is limited, we can only run additional experiments with NGU in Minigrid tasks. In this revision, NGU's result is clearly shown as inferior to our method (see Table 1).

---

### Official Review · Reviewer_Qsrw · 2022-11-04

**Confidence:** 4
**Correctness:** 3
**Technical Novelty And Significance:** 2
**Empirical Novelty And Significance:** 2
**Recommendation:** 5

**Clarity, Quality, Novelty And Reproducibility:**

Novelty: Intrinsic motivation in reinforcement learning is an active area of research. "Surprise" is a concept that has been used frequently in this literature, and so is "novelty". The current paper builds on this literature to propose a new source of intrinsic motivation, namely the novelty of the surprise, as well as a mechanism for computing a precise value of intrinsic reward .

Clarity: The writing can be much improved. The main paper is repetitive and takes a long time to get to the point and to precise definitions. A large number of acronyms are introduced, which creates unnecessary difficulty for the reader. Page 1 is not the ideal placement for Figure 1, especially in its current form (see further comments on this below). Some important pieces of information are not provided in the main paper (e.g., description of the domains, which is critical for understanding the results). Some important results are relegated to the Appendices, which necessitated a lot of back-and-forth between the main paper and the appendices when reading the paper. Performance metrics should be clearly and explicitly defined in the main paper (e.g., In Figure 5, what precisely is being plotted?).

The six frames and associated intrinsic rewards in Figure 1 are difficult to understand and evaluate without context. A much more useful figure would show the complete history of rewards through one or more episodes, along with associated frames at certain decision stages (see, for example, Figure 1 by Burda et al, 2018).

Similarly, in Figure 3, intrinsic and extrinsic rewards can be shown as the agent is interacting with the environment. Currently, the figure shows 7 sample frames, which I did not find to be particularly informative without further context.

Learning curves corresponding to Table 1 would be useful to see in the main paper.

Quality: The empirical evaluation is extensive but it would be useful to see further interrogation in some areas. One question that arises is the generality of the experimental results to other domains. Much of the discussion and results in the paper are on visual domains. In addition, the domains appear to be either deterministic or stochastic in a particular way (e.g., the stochasticity in the noisy-TV domain is entirely irrelevant to the task.) Another area that deserves further interrogation is the structure of the Surprise Memory. It would be useful to see an exploration of the design choices made here (ideally, with some alternative approaches). The ablation study in section 4 is on a single domain and has a narrow scope. Finally, while I appreciate the existing efforts of the authors to illustrate the behavior of the algorithm (e.g., Figure 4), the main emphasis in the paper is on high-level performance comparison (e.g., total reward obtained), with the result that the proposed computation of "surprise novelty" is not deeply understood.


**Strength And Weaknesses:**

Strengths: The paper is in an important area of research relevant to the conference. The proposed concept for intrinsic reward is intriguing and can be useful. There is extensive empirical analysis, showing improved performance compared to existing approaches.

Weakness: Experimental analysis has some limitations (see below) with the result that the general applicability and benefits of the approach are difficult to evaluate. The writing is relatively poor.



**Summary Of The Paper:**

The authors propose a new intrinsic reward function for reinforcement learning, building on earlier definitions of "surprise". The proposed idea is to reward not the surprise itself but its novelty. The authors propose a specific approach to computing this intrinsic reward function and present an empirical analysis in a number of domains.

**Summary Of The Review:**

The paper presents a potentially useful concept for intrinsic reward, as well as a particular implementation of it. But I struggled to reach a non-superficial understanding of the behaviour of the proposed method and to evaluate its general applicability and usefulness.

---

> ### Author Response · Authors · 2022-11-16
> **Reply to Reviewer Qsrw**
>
> Thank you for your thoughtful review. We address your concern point by point as follows.
>
> >"Some important pieces of information are not provided in the main paper (e.g., description of the domains, which is critical for understanding the results)."
>
> We described the characteristics of the domains for our Noisy-TV and 3 Minigrid tasks in Sec. 3.1 and 3.2, respectively. We only skipped the description of Atari games, which is already well-known in the community. In the revision, we have added more information in Appendix D.
>
> > "Some important results are relegated to the Appendices, which necessitated a lot of back-and-forth between the main paper and the appendices when reading the paper. "
>
> We reported all performance results in the main text. The Appendix only reports detailed learning curves and additional behaviour analysis. We will move some of the learning curves to the main text if more space is allowed.
>
> > "Performance metrics should be clearly and explicitly defined in the main paper (e.g., In Figure 5, what precisely is being plotted?)."
>
> Thank you for your advice. The plotted metric is the average episodic return taken over 128 testing episodes. It was mentioned briefly in Appendix D3. We have noted that explicitly in the revision's captions.
>
> > Fig. 1
>
> As mentioned in the caption, Fig.1 shows summarized statistics of the surprise novelty and norm. The numbers are taken on average for each room, and the six figures are sampled images representing each room. We do not intend to show the metric for a specific frame, which is unreliable. For complicated games like Montezuma, we believe it is easier for readers to compare the originality on room level rather than frame level (e.g., black room is less attractive than ladder room). Hence, we do not want to follow  Burda et al., 2018.
>
> > Fig. 3
>
> We agree with you and have changed the figure in this revision, showing the surprise novelty and surprise norm over interaction steps in an episode. The 7 figures are sampled from these steps.
>
> > "Learning curves corresponding to Table 1"
>
> Due to space constraints, we leave learning curves in Appendix. When more space is available, we will add them to the main text as requested.
>
> > "the generality of the experimental results to other domains"
>
> We want to clarify that our experiment already covered three different domains:
> - Noisy-TV: 3d navigation. The state is an image of the viewport.
> - MiniGrid: grid-world environment. The state is a description vector representing partial observation information such as the location of the agent, objects, moving directions, etc.
> - Atari games. The state is a 2d image representing the screen of the game.
>
> We have added this information to the Appendix of the new manuscript. We believe that our experimented domains are diverse compared to prior works (e.g., Burda et al. (2018))
>
> > "Another area that deserves further interrogation is the structure of the Surprise Memory."
>
> Thank you for your advice. We have already conducted an ablation study on the SM component for MiniGrid's DO and Atari's Montezuma Revenge. We also compared SM with the count-based approach in Noisy-TV.
>
> Due to computing constraints, we could not run ablation studies for all tasks. This revision has added more ablation studies on Frostbite and Venture in Appendix Fig. 10, confirming the necessity of M and W.  We hope that it will clear your concern.

---

### Author Response · Authors · 2022-11-16
**General response**

Dear Reviewers,

We appreciate your effort in reviewing our paper. We are glad that the reviewers find:
- Our method is "intriguing, interesting" (Reviewer Qsrw and ZXKP), "novel, well-motivated and non-trivial" (Reviewer 422A and oPBQ), "integrated easily on existing methods" (Reviewer 422A)
- Our paper has an "extensive empirical analysis" (Reviewer Qsrw), an "adequate amount of experimental analysis" (Reviewer ZXKP) and "abundant theoretical and experimental ground" (Reviewer oPBQ).
- Our writing is "clear" and "well presented and easy to follow" (Reviewer 422A and Reviewer oPBQ)

However, there remain limitations, misunderstandings and questions. We will address these by replying to each of your reviews. Please consider increasing your score if you find our responses valid.

---

### Comment · Area_Chair_c8uq · 2022-11-18
**Responses**

Dear Reviewers,

Do you have any comments/replies to author's responses - it would be great if you could respond to them. Have they changed your opinion on the paper?

Kind regards,
AC

---

### Comment · Area_Chair_c8uq · 2022-11-22
**Couple of comments**

Dear Authors

Couple of comments from me: I would also like simple clear explanation the method makes sense. In particular, I don't understand why is the auto encoder is reconstructing q=[u^e_t,u_t] (concatenation?) - why this combination? And why the difference between the actual value and its reconstruction from AE is the surprise novelty. For example, I would expect surprise novelty to be something like difference between retrieved and actual value of the prediction error u.
In terms of model presentation, it is slightly hard to read from the text, but the Figure 2 is quite clear. Perhaps making it more self contained would be helpful. You could consider: showing how u is obtained (how SG works - or one example of it that you are using). Also the r is the norm of the difference which you are not showing and how q is constructed from the u's and perhaps add word "memory retrieval" for obtaining ue.

Thank you,
AC

---

> ### Author Response · Authors · 2022-11-25
> **Reply to Area Chair**
>
> We thank the AC for your constructive comments. We will answer your questions below.
>
> > why is the auto encoder is reconstructing q=[u^e_t,u_t] (concatenation?) - why this combination?
>
> The motivation for including $u^e_t$ in the reconstruction process is given in the third paragraph of the introduction (A further mechanism ...). We then provide a clear example of the importance of reconstructing both $q=[u^e_t,u_t]$ instead of $q=u_t$ in the second paragraph of Sec. 2.3.
>
> In essence, $u_t$ represents long-term inter-episode surprises while $u^e_t$ is short-term intra-episode surprises. We need both to capture a comprehensive representation of surprise. Without $u^e_t$, when the SG and W are fixed (during interaction with environments), the outputs $u_t$ and $r^i_t$ stay the same for the same input $I_t$. It is undesirable since when the agent observes the same input at different timesteps (e.g., $I_1 = I_2$), we expect its curiosity should decrease in the second visit ($r^i_1<r^i_2$), it cannot be achieved with only $u_t$ .
>
> > In terms of model presentation, it is slightly hard to read from the text, but the Figure 2 is quite clear.
>
> Thank you for your advice. We will update the figure as suggested in the next revision of our manuscript.

---

### Decision · Program_Chairs · 2023-01-20

**Decision:**

Reject

**Justification For Why Not Higher Score:**

The motivation is unclear, and either needs better explanation or more extensive experiments.

**Justification For Why Not Lower Score:**

The method works on domains tested.

**Metareview: Summary, Strengths And Weaknesses:**

This method introduces a specific intrinsic reward for exploration and achieves improvements over methods it extends on small number of domains. However, at the first glance, this reward looks somewhat arbitrary and therefore needs justification. Despite authors giving one, several reviewers had hard time understanding the method or why it should work (I also have hard time following it). I recommend that either the authors work more on explaining the method or provide more extensive experimental evaluation justifying that the method works in general, regardless if it easy or hard to understand why.